# FINE Samples for Learning with Noisy Labels

**Taehyeon Kim** [*]
KAIST AI
KAIST
Daejeon, South Korea
potter32@kaist.ac.kr

**Jongwoo Ko** [*]
KAIST AI
KAIST
Daejeon, South Korea
jongwoo.ko@kaist.ac.kr

**Sangwook Cho**
KAIST AI
KAIST
Daejeon, South Korea
sangwookcho@kaist.ac.kr

**Jinhwan Choi**
KAIST AI
KAIST
Daejeon, South Korea
jinhwanchoi@kaist.ac.kr

**Se-Young Yun**
KAIST AI
KAIST
Daejeon, South Korea
yunseyoung@kaist.ac.kr

## Abstract

Modern deep neural networks (DNNs) become weak when the datasets contain noisy (incorrect) class labels. Robust techniques in the presence of noisy labels can be categorized into two types: developing *noise-robust* functions or using *noise-cleansing* methods by detecting the noisy data. Recently, *noise-cleansing* methods have been considered as the most competitive noisy-label learning algorithms. Despite their success, their noisy label detectors are often based on heuristics more than a theory, requiring a robust classifier to predict the noisy data with loss values. In this paper, we propose a novel detector for filtering label noise. Unlike most existing methods, we focus on each data point's latent representation dynamics and measure the alignment between the latent distribution and each representation using the eigen decomposition of the data gram matrix. Our framework, coined as *filtering noisy instances via their eigenvectors* (FINE), provides a robust detector using derivative-free simple methods with theoretical guarantees. Under our framework, we propose three applications of the FINE: sample-selection approach, semi-supervised learning (SSL) approach, and collaboration with *noise-robust* loss functions. Experimental results show that the proposed methods consistently outperform corresponding baselines for all three applications on various benchmark datasets [1].

## 1 Introduction

Deep neural networks (DNNs) have achieved remarkable success in numerous tasks as the amount of accessible data has dramatically increased [21, 15]. On the other hand, accumulated datasets are typically labeled by a human, a labor-intensive job or through web crawling [48] so that they may be easily corrupted (*label noise*) in real-world situations. Recent studies have shown that deep neural networks have the capacity to memorize essentially any labeling of the data [49]. Even a small amount of such noisy data can hinder the generalization of DNNs owing to their strong memorization of noisy labels [49, 29]. Hence, it becomes crucial to train DNNs that are robust to corrupted labels. As label noise problems may appear anywhere, such robustness increases reliability in many applications such as the e-commerce market [9], medical fields [45], on-device AI [46], and autonomous driving systems [11].

To improve the robustness against noisy data, the methods for learning with noisy labels (LNL) have been evolving in two main directions [18]: (1) designing noise-robust objective functions or regular-

---

[*]Equal contribution
[1]Code available at `https://github.com/Kthyeon/FINE_official`

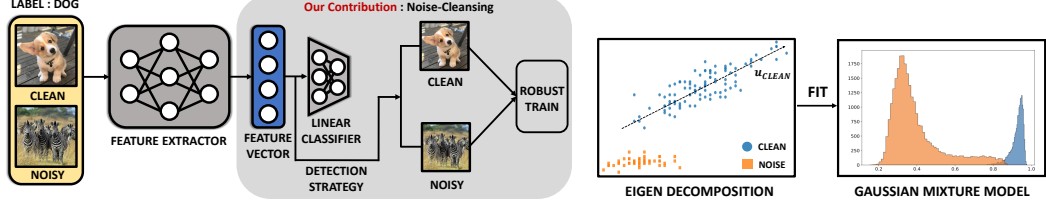

(a) Noise-Cleansing-based Approach            (b) FINE

Figure 1: Illustration of (a) basic concept of this work and (b) proposed detection framework, FINE. Noise-cleansing learning generally separates clean data from the original dataset by using prediction outputs. We propose a novel derivative-free detector based on an unsupervised clustering algorithm on the high-order topological space. FINE measures the alignment of pre-logits (i.e., penultimate layer representation vectors) toward the class-representative vector that is extracted through the eigen decomposition of the gram matrix of data representations.

izations and (2) detecting and cleansing the noisy data. In general, the former *noise-robust* direction uses explicit regularization techniques [6, 52, 50] or robust loss functions [38, 13, 40, 51], but their performance is far from state-of-the-art [49, 26] on datasets with severe noise rates. Recently, researchers have designed *noise-cleansing* algorithms focused on segregating the clean data (i.e., samples with uncorrupted labels) from the corrupted data [19, 14, 47, 18, 32, 42]. One of the popular criteria for the segregation process is the loss value between the prediction of the noisy classifier and its noisy label, where it is generally assumed that the noisy data have a large loss [19, 14, 47, 18] or the magnitude of the gradient during training [51, 40]. However, these methods may still be biased by the corrupted linear classifier towards label noise because their criterion (e.g., loss values or weight gradient) uses the posterior information of such a linear classifier [24]. Maennel et al. [31] analytically showed that the principal components of the weights of a neural network align with the randomly labeled data; this phenomenon can yield more negative effects on the classifier as the number of randomly labeled classes increases. Recently, Wu et al. [42] used an inherent geometric structure induced by nearest neighbors (NN) in latent space and filtered out isolated data in such topology, and its quality was sensitive to its hyperparameters regarding NN clustering in the presence of severe noise rates.

To mitigate such issues for label noise detectors, we provide a novel yet simple detector framework, *filtering noisy labels via their eigenvectors* (FINE) with theoretical guarantees to provide a high-quality splitting of clean and corrupted examples (without the need to estimate noise rates). Instead of using the neural network's linear classifier, FINE utilizes the principal components of latent representations made by eigen decomposition which is one of the most widely used unsupervised learning algorithms and separates clean data and noisy data by these components (Figure 1a). To motivate our approach, as Figure 1b shows, we find that the clean data (blue points) are mainly aligned on the principal component (black dotted line), whereas the noisy data (orange points) are not; thus, the dataset is well clustered with the alignment of representations toward the principal component by fitting them into Gaussian mixture models (GMM). We apply our framework to various *LNL* methods: the sample selection approach, a semi-supervised learning (SSL) approach, and collaboration with noise-robust loss functions. The key contributions of this work are summarized as follows:

- We propose a novel framework, termed FINE (*filtering noisy labels via their eigenvectors*), for detecting clean instances from noisy datasets. FINE makes robust decision boundary for the high-order topological information of data in latent space by using eigen decomposition of their gram matrix.

- We provide provable evidence that FINE allows a meaningful decision boundary made by eigenvectors in latent space. We support our theoretical analysis with various experimental results regarding the characteristics of the principal components extracted by our FINE detector.

- We develop a simple sample-selection method by replacing the existing detector method with FINE. We empirically validate that a sample-selection learning with FINE provides consistently superior detection quality and higher test accuracy than other existing alternative methods such as the Co-teaching family [14, 47], TopoFilter [42], and CRUST [32].

- We experimentally show that our detection framework can be applied in various ways to existing *LNL* methods and validate that ours consistently improves the generalization in the presence of noisy data: sample-selection approach [14, 47], SSL approach [25], and collaboration with noise-robust loss functions [51, 40, 29].

**Organization.** The remainder of this paper is organized as follows. In Section 2, we discuss the recent literature on LNL solutions and meaningful detectors. In Section 3, we address our motivation for creating a noisy label detector with theoretical insights and provide our main method, filtering the noisy labels via their eigenvectors (FINE). In Section 4, we present the experimental results. Finally, Section 5 concludes the paper.

## 2   Related Works

Zhang et al. [49] empirically showed that any convolutional neural networks trained using stochastic gradient methods easily fit a random labeling of the training data. To tackle this issue, numerous works have examined the classification task with noisy labels. We do not consider the works that assumed the availability of small subsets of training data with clean labels [17, 36, 39, 53, 3].

**Noise-Cleansing-based Approaches.** Noise-cleansing methods have evolved following the improvement of noisy detectors. Han et al. [14] suggested a noisy detection approach, named co-teaching, that utilizes two networks, extracts subsets of instances with small losses from each network, and trains each network with subsets of instances filtered by another network. Yu et al. [47] combined a disagreement training procedure with co-teaching, which only selects instances predicted differently by two networks. Huang et al. [18] provided a simple noise-cleansing framework, training-filtering-training; the empirical efficacy was improved by first finding label errors, then training the model only on data predicted as clean. Recently, new noisy detectors with theoretical support have been developed. Wu et al. [42] proposed a method called TopoFilter that filters noisy data by utilizing the k-nearest neighborhood algorithm and Euclidean distance between pre-logits. Mirzasoleiman et al. [32] introduced an algorithm that selects subsets of clean instances that provide an approximately low-rank Jacobian matrix and proved that gradient descent applied to the subsets prevents overfitting to noisy labels. Pleiss et al. [34] proposed an area under margin (AUM) statistic that measures the average difference between the logit values of the assigned class and its highest non-assigned class to divide clean and noisy samples. Cheng et. al [8] progressively filtered out corrupted instances using a novel confidence regularization term. The noise-cleansing method was also developed in a semi-supervised learning (SSL) manner. Li et al. [25] modeled the per-sample loss distribution and divide it into a labeled set with clean samples and an unlabeled set with noisy samples, and they leverage the noisy samples through the well-known SSL technique MixMatch [4].

**Noise-Robust Models.** Noise-robust models have been studied in the following directions: robust-loss functions, regularizations, and strategies. First, for robust-loss functions, Ghosh et al. [13] showed that the mean absolute error (MAE) might be robust against noisy labels. Zhang & Sabuncu et al. [51] argued that MAE performed poorly with DNNs and proposed a GCE loss function, which can be seen as a generalization of MAE and cross-entropy (CE). Wang et al. [40] introduced the reverse version of the cross-entropy term (RCE) and suggested that the SCE loss function is a weighted sum of the CE and RCE. Some studies have stated that the early-stopped model can prevent the memorization phenomenon for noisy labels [2, 49] and theoretically analyzed it [26]. Based on this hypothesis, Liu et al. [29] proposed an early-learning regularization (ELR) loss function to prohibit memorizing noisy data by leveraging the semi-supervised learning techniques. Xia et al. [43] clarified which neural network parameters cause memorization and proposed a robust training strategy for these parameters. Efforts have been made to develop regularizations on the prediction level by smoothing the one-hot vector [30], using linear interpolation between data instances [50], and distilling the rescaled prediction of other models [20]. However, these works have limitations in terms of performance as the noise rate of the dataset increases.

**Dataset Resampling.** Label-noise detection may be a category of data resampling which is a common technique in the machine learning community that extracts a "*helpful*" dataset from the distribution of the original dataset to remove the dataset bias. In class-imbalance tasks, numerous studies have conducted over-sampling of minority classes [7, 1] or undersampling the majority classes [5] to balance the amount of data per class. Li & Vasconcelos et al. [27] proposed a resampling procedure to reduce the representation bias of the data by learning a weight distribution that favors difficult instances for a given feature representation. Le Bras et al. [22] suggested an adversarial filtering-based approach to remove spurious artifacts in a dataset. Analogously, in anomaly detection and

---

**Algorithm 1:** FINE Algorithm for Sample Selection

---

INPUT : Noisy training data $\mathcal{D}$, feature extractor $g$, number of classes $K$, clean probability threshold $\zeta$, set of FINE scores for class $k$ $\mathcal{F}_k$

OUTPUT : Collected clean data $\mathcal{C}$

1: Initialize $\mathcal{C} \leftarrow \emptyset$, $\hat{\mathcal{D}} \leftarrow \mathcal{D}$, $\boldsymbol{\Sigma}_k \leftarrow \mathbf{0}$ for all $k = 1, \ldots, K$
   /* Update the convariance matrices for all classes */
2: **for** $(\boldsymbol{x}_i, y_i) \in \mathcal{D}$ **do**
3:    $\boldsymbol{z}_i \leftarrow g(\boldsymbol{x}_i)$
4:    Update the gram matrix $\boldsymbol{\Sigma}_{y_i} \leftarrow \boldsymbol{\Sigma}_{y_i} + \boldsymbol{z}_i \boldsymbol{z}_i^\top$
5: **end for**
   /* Generate the principal component with eigen decomposition */
6: **for** $k = 1, \ldots, K$ **do**
7:    $\mathbf{U}_k, \boldsymbol{\Lambda}_k \leftarrow$ EIGEN DECOMPOSITION OF $\boldsymbol{\Sigma}_k$
8:    $\mathbf{u}_k \leftarrow$ THE FIRST COLUMN OF $\mathbf{U}_k$
9: **end for**
   /* Compute the alignment score and get clean subset $\mathcal{C}$ */
10: **for** $(\boldsymbol{x}_i, y_i) \in \mathcal{D}$ **do**
11:    Compute the FINE score $f_i = \langle \mathbf{u}_{y_i}, \boldsymbol{z}_i \rangle^2$ and $\mathcal{F}_{y_i} \leftarrow \mathcal{F}_{y_i} \cup \{f_i\}$
12: **end for**
   /* Finding the samples whose clean probability is larger than $\zeta$ */
13: $\mathcal{C} \leftarrow \mathcal{C} \cup \text{GMM}(\mathcal{F}_k, \zeta)$ for all $k = 1, \ldots, K$

---

out-of-distribution detection problems [16, 28, 23], the malicious data are usually detected by examining the loss value or negative behavior in the feature representation space. While our research is motivated by such previous works, this paper focuses on the noisy image classification task.

## 3 Method

In this section, we present our detector framework and the theoretical motivation behind using the detector in high-dimensional classification. To segregate the clean data, we utilize the degree of alignment between the representations and the eigenvector of the representations' gram matrices for all classes, called **FINE** (*FIltering Noisy instnaces via their Eigenvectors*). Our algorithm is as follows (Algorithm 1). FINE first creates a gram matrix of the representation in the noisy training dataset for each class and conducts the eigen decomposition for those gram matrices. Then, FINE finds clean and noisy instances using the square of inner product values between the representations and the first eigenvector having the largest eigenvalue. In this manner, we treat the data as clean if aligned onto the first eigenvector, while most of the noisy instances are not. Here, we formally define '*alignment*' and '*alignment clusterability*' in Definition 1 and Definition 2, respectively.

**Definition 1.** *(Alignment) D*

**Definition 2.** *(Alignment Clusterability) For all features labeled as class k in dataset $\mathcal{D}$, let fit a Gaussian Mixture Model (GMM) on their alignment (Definition 1) distribution to divide current samples into a clean set and a noisy set; the set having larger mean value is treated as a clean set, and another one is a noisy set. Then, we say a dataset $\mathcal{D}$ satisfies alignment clusterability if the representation $\boldsymbol{z}$ labeled as the same true class belongs to the clean set.*

As an empirical evaluation, the quality of our detector for noisy data is measured with the ***F-score***, a widely used criterion in noisy label detection, anomaly detection and out-of-distribution detection [8, 16, 28, 23]. We treat the selected clean samples as the positive class and the noisy samples as negative class. The ***F-score*** is the harmonic mean of the precision and the recall; the *precision* indicates the fraction of clean samples among all samples that are predicted as clean, and the *recall* indicates the portion of clean samples that are identified correctly.

### 3.1 Alignment Analysis for Noisy Label Detector

To design a robust label noise filtering framework, we explore the linear nature of the topological space of feature vectors for data resampling techniques and deal with the classifier contamination due to random labels. Recent studies on the distribution of latent representations in DNNs provide

insight regarding how correctly the outlier samples can be filtered with the hidden space's geometrical information. For instance, in [23, 24], the authors proposed frameworks for novelty detection using the topological information of pre-logit based on the Mahalanobis distance, and, in [42], the authors filtered the noisy data based on the Euclidean distance between pre-logits. Maennel et al. [31] analytically showed that an alignment between the principal components of network parameters and those of data takes place when training with random labels. This finding points out that random labels can corrupt a classifier, and thus building a robust classifier is required.

Motivated by these works, we aim to design a novel detector using the principal components of latent features to satisfy Definition 2. However, it is intractable to find the optimal classifier to maximize the separation of *alignment clusterability* because clean data distribution and noisy data distribution are inaccessible. To handle this issue, we attempt to approximate the clean eigenvector to maximize the alignment values of clean data rather than to maximize the separation; the algorithm utilizes the eigenvector of the data for each class (Figure 2). Below, we provide the upper bound for the perturbation toward the clean data's eigenvector under simple problem settings with noisy labels referred to in other studies [29, 42]. We first introduce notations. Next, we establish the theoretical evidence that our FINE algorithm approximates the clean data's eigenvectors under some assumptions for its analytic tractability (Theorem 1). We mainly present the theorem and its interpretation; details of the proofs can be found in the Appendix.

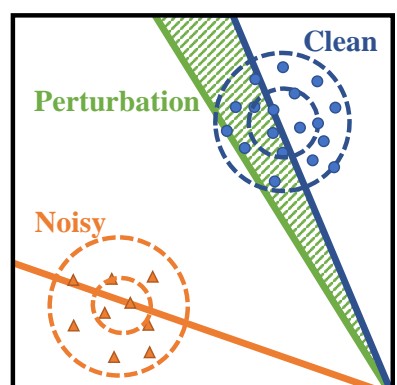

Figure 2: Illustration for the problem settings and Theorem 1. The perturbation (green shade) is the angle between the first eigenvector of clean instances (blue line) and the estimated first eigenvector (green line) which is perturbed by that of noisy instances (orange line). Note that blue and orange points are clean instances and noisy instances, respectively.

**Notations.** Consider a binary classification task. Assume that the data points and labels lie in $\mathcal{X} \times \mathcal{Y}$, where the feature space $\mathcal{X} \subset \mathbb{R}^d$ and label space $\mathcal{Y} = \{-1, +1\}$. A single data point $\boldsymbol{x}$ and its true label $y$ follow a distribution $(\boldsymbol{x}, y) \sim P_{\mathcal{X} \times \mathcal{Y}}$. Denote by $\tilde{y}$ the observed label (potentially corrupted). Without loss of generality, we focus on the set of data points whose observed label is $\tilde{y} = +1$.

Let $\mathbf{X} \subset \mathcal{X}$ be the finite set of features with clean instances whose true label is $y = +1$. Similarly, let $\tilde{\mathbf{X}} \subset \mathcal{X}$ be the set of noisy instances whose true label is $y = -1$. To establish our theorem, we assume the following reasonable conditions referred to other works using linear discriminant analysis (LDA) assumptions [24, 12]:

**Assumption 1.** *The feature distribution is comprised of two Gaussians, each identified as a clean cluster and a noisy cluster.*

**Assumption 2.** *The features of all instances with $y = +1$ are aligned on the unit vector $\mathbf{v}$ with the white noise, i.e., $\mathbb{E}_{\boldsymbol{x} \in \mathbf{X}}[\boldsymbol{x}] = \boldsymbol{v}$. Similarly, features of all instances with $y = -1$ are aligned on the unit vector $\boldsymbol{w}$, i.e., $\mathbb{E}_{\boldsymbol{x} \in \tilde{\mathbf{X}}}[\boldsymbol{x}] = \boldsymbol{w}$.*

**Theorem 1.** *(Upper bound for the perturbation towards the clean data's eigenvector $\boldsymbol{v}$) Let $N_+$ and $N_-$ be the number of clean instances and noisy instances, respectively, and $\boldsymbol{u}$ be the FINE's eigenvector which is the first column of $\mathbf{U}$ from the eigen decomposition of the whole data's matrix $\boldsymbol{\Sigma}$. For any $\delta \in (0, 1)$, its perturbation towards the $\boldsymbol{v}$ in assumption 2 (i.e., 2-norm for difference of projection matrices; left hand side of Eq. (1)) holds the following with probability $1 - \delta$:*

$$\left\| \boldsymbol{u}\boldsymbol{u}^\top - \boldsymbol{v}\boldsymbol{v}^\top \right\|_2 \leq \frac{3\tau \cos\theta + \mathcal{O}(\sigma^2 \sqrt{\frac{d + \log(4/\delta)}{N_+}})}{1 - \tau(\sin\theta + 3\cos\theta) - \mathcal{O}(\sigma^2 \sqrt{\frac{d + \log(4/\delta)}{N_+}})} \tag{1}$$

*where $\boldsymbol{w}$ is the first eigenvector of noisy instances, $\tau$ is the fraction between noisy and clean instances ($\frac{N_-}{N_+}$), $\theta$ is $\angle(\boldsymbol{w}, \boldsymbol{v})$, and $\sigma^2$ is a variance of white noise.*

Theorem 1 states that the upper bound for the perturbation toward $\boldsymbol{v}$ are dependent on both the ratio $\tau$ and the angle $\theta$ between $\boldsymbol{w}$ and $\boldsymbol{v}$; small upper bound can be guaranteed as the number of clean data

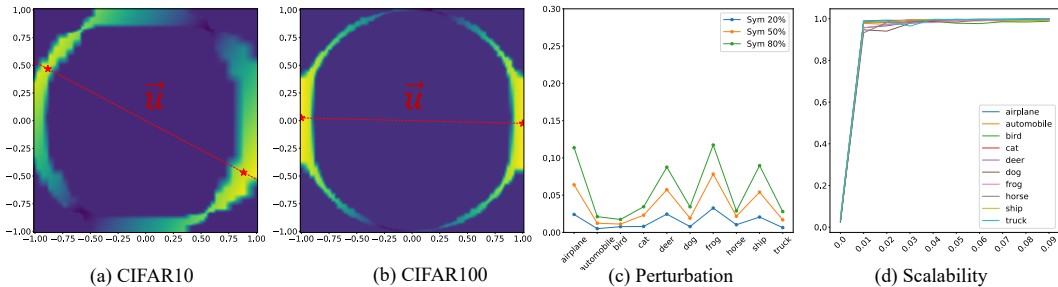

| (a) CIFAR10 | (b) CIFAR100 | (c) Perturbation | (d) Scalability |

Figure 3: (a), (b): Heatmaps of Eq. (2) values on unit circle in random hyperplane. We evaluate this visualization on the ResNet34 model trained with common cross-entropy loss on CIFAR-10 with asymmetric noise 40% and CIFAR-100 with symmetric noise 80%, respectively. Colors closer to yellow indicate larger the values; (c): comparison of perturbations of Eq. (1) on CIFAR-10 with symmetric noise 20%; (d): comparison of cosine similarity values between FINE's principal components and approximated principal components using fraction of data on CIFAR-10 with symmetric noise 80%.

increases, $\tau$ decreases, and $\theta$ approaches $\frac{\pi}{2}$. We also derive the lower bound for the precision and the recall when using the eigenvector $u$ in Appendix. In this theoretical analysis, we can ensure that such lower bound values become larger as $v$ and $w$ become orthogonal to each other. To verify these assumptions, we provide various experimental results for the separation of *alignment clusterability*, the perturbation values, the scalability to the number of samples, the quality of our detector in the application of sample selection approach, and the comparison with an alternative clustering-based estimator [24].

**Validation for our Estimated Eigenvector.**  To validate our FINE's principal components, we first propose a simple visualization scheme based on the following steps: (1) Pick the first eigenvector ($u$) extracted by FINE algorithm, (2) Generate a random hyperplane spanned by such eigenvector ($u$) and a random vector, (3) Calculate the value of the following Eq. (2) on any unit vectors ($a$) in such hyperplane and plot a heatmap with them:

$$\frac{1}{|\mathbf{X}|} \sum_{\boldsymbol{x}_i \in \mathbf{X}} \langle \boldsymbol{a}, \boldsymbol{x}_i \rangle^2 - \frac{1}{|\tilde{\mathbf{X}}|} \sum_{\boldsymbol{x}_j \in \tilde{\mathbf{X}}} \langle \boldsymbol{a}, \boldsymbol{x}_j \rangle^2 \qquad (2)$$

Eq. (2) is maximized when the unit vector $a$ not only maximizes the FINE scores of clean data for the first term in Eq. (2), but also minimizes those of noisy data for the second term in Eq. (2). This visualization shows in 2-D how FINE's first eigenvector ($u$) optimizes such values in the presence of noisy instances (Figure 3a and 3b). As the figures show, the FINE's eigenvector $u$ (red dotted line) has almost maximum value of Eq. (2). Furthermore, we empirically evaluate the perturbation values in Theorem 1 as the noise rate changes (Figure 3c); FINE has small perturbation values even in a severe noise rate.

**Scalability to Number of Samples.**  Despite FINE's superiority, it may require high computational costs if the whole dataset is used for eigen decomposition. To address this issue, we approximate the eigenvector with a small portion of the dataset and measure the cosine similarity values between the approximated term and the original one ($u$) (Figure 3d). Interestingly, we verify that far accurate eigenvector is computed even using 1% data (i.e., a cosine similarity value is 0.99), and thus the eigenvector can be accurately estimated with little computation time.

**Validation for Dynamics of Sample-selection Approach.**  We evaluate the F-score dynamics of every training epoch on the symmetric and the asymmetric label noise in Figure 4. We compare FINE with the following sample-selection approaches: Co-teaching [14] and TopoFilter [42]. In Figure 4, during the training process, F-scores of FINE becomes consistently higher on both symmetric noise and asymmetric noise settings, while Co-teaching and TopoFilter achieve lower quality. Unlike TopoFilter and FINE, Co-teaching even performs the sample-selection with the access of noise rate. This evidence show that FINE is also applicable to the naive sample-selection approaches.

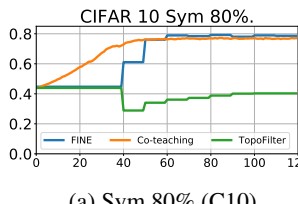 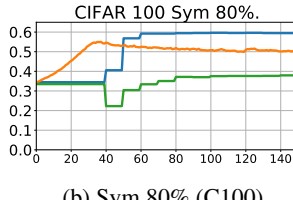 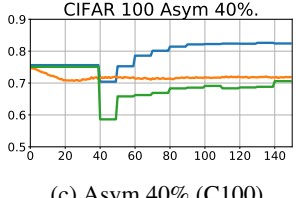

| (a) Sym 80% (C10) | (b) Sym 80% (C100) | (c) Asym 40% (C100) |

Figure 4: Comparisons of F-scores on CIFAR10 and CIFAR100 under symmetric and asymmetric label noise. C10 and C100 denote CIFAR-10 and CIFAR-100, respectively.

**Comparison for Mahalanobis Distance Estimator** Under similar conditions, Lee et al.[24] measured the Mahalanobis distance of pre-logits using the minimum covariance determinant(MCD) estimator and selected clean samples based on this distance. While they also utilized the LDA assumptions on pre-logits, FINE consistently outperforms MCD in both precision and recall, thus yielding better F-score(Figure 5). The experimental results justify our proposed detector, in comparison with a similar alternative.

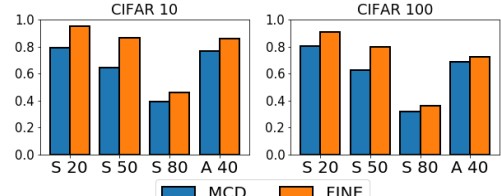

Figure 5: Comparisons of F-scores on CIFAR-10 and CIFAR-100 under symmetric (S) and asymmetric noise (A) settings.

## 4 Experiment

In this section, we demonstrate the effectiveness of our FINE detector for three applications: sample selection approach, SSL, and collaboration with noise-robust loss functions.

### 4.1 Experimental Settings

**Noisy Benchmark Dataset.** Following the previous setups [25, 29], we artificially generate two types of random noisy labels: injecting uniform randomness into a fraction of labels (symmetric) and corrupting a label only to a specific class (asymmetric). For example, we generate noise by mapping TRUCK → AUTOMOBILE, BIRD → AIRPLANE, DEER → HORSE, CAT ↔ DOG to make asymmetric noise for CIFAR-10. For CIFAR-100, we create 20 five-size super-classes and generate asymmetric noise by changing each class to the next class within super-classes circularly. For a real-world dataset, Clothing1M [44] containing inherent noisy labels is used. This dataset contains 1 million clothing images obtained from online shopping websites with 14 classes[2]. The dataset provides $50k, 14k$, and $10k$ verified as clean data for training, validation, and testing. Instead of using the $50k$ clean training data, we use a randomly sampled pseudo-balanced subset as a training set with $120k$ images. For evaluation, we compute the classification accuracy on the $10k$ clean dataset.

**Networks and Hyperparameter Settings.** We use the architectures and hyperparameter settings for all baseline experiments following the setup of Liu et al.[29] except with SSL approaches. For SSL approaches, we follow the setup of Li et al.[25]. We set the threshold $\zeta$ as $0.5$.

### 4.2 Application of FINE

#### 4.2.1 Sample Selection-Based Approaches

We apply our FINE detector for various sample selection algorithms. In detail, after warmup training, at every epoch, FINE selects the clean data with the eigenvectors generated from the gram matrices of data predicted to be clean in the previous round, and then the neural networks are trained with them. We compare our proposed method with the following sample selection approaches: (1) *Bootstrap* [35], (2) *Forward* [33], (3) *Co-teaching* [14]; (4) *Co-teaching+* [47]; (5) *TopoFilter* [42]; (6) *CRUST* [32]. We evaluate these algorithms three times and report error bars.

---

[2]T-shirt, Shirt, Knitwear, Chiffon, Sweater, Hoodie, Windbreaker, Jacket, Down Coat, Suit, Shawl, Dress, Vest, and Underwear. The labels in this dataset are extremely noisy (estimated noisy level is 38.5%) [37].

Table 1: Test accuracies (%) on CIFAR-10 and CIFAR-100 under different noisy types and fractions. All comparison methods are reproduced with publicly available code, while the results for Bootstrap [35] and Forward [33] are taken from [29]. For CRUST [32], we experiment without mix-up to compare the intrinsic sample selection effect of each method. The average accuracies and standard deviations over three trials are reported. Here, we substitute the sample selection method of Co-teaching [14, 47] with FINE (i.e., F-Co-teaching). The best results sharing the noisy fraction and method are highlighted in bold.

| Dataset | CIFAR-10 | | | | CIFAR-100 | | | |
|---|---|---|---|---|---|---|---|---|
| Noisy Type | Sym | | | Asym | Sym | | | Asym |
| Noise Ratio | 20 | 50 | 80 | 40 | 20 | 50 | 80 | 40 |
| Standard | $87.0 \pm 0.1$ | $78.2 \pm 0.8$ | $53.8 \pm 1.0$ | $85.0 \pm 0.0$ | $58.7 \pm 0.3$ | $42.5 \pm 0.3$ | $18.1 \pm 0.8$ | $42.7 \pm 0.6$ |
| Bootstrap [35] | $86.2 \pm 0.2$ | - | $54.1 \pm 1.3$ | $81.2 \pm 1.5$ | $58.3 \pm 0.2$ | - | $21.6 \pm 1.0$ | $45.1 \pm 0.6$ |
| Forward [33] | $88.0 \pm 0.4$ | - | $54.6 \pm 0.4$ | $83.6 \pm 0.6$ | $39.2 \pm 2.6$ | - | $9.0 \pm 0.6$ | $34.4 \pm 1.9$ |
| Co-teaching [14] | $89.3 \pm 0.3$ | $83.3 \pm 0.6$ | $66.3 \pm 1.5$ | $88.4 \pm 2.8$ | $63.4 \pm 0.0$ | $49.1 \pm 0.4$ | $20.5 \pm 1.3$ | $47.7 \pm 1.2$ |
| Co-teaching+ [47] | $89.1 \pm 0.5$ | $84.9 \pm 0.4$ | $63.8 \pm 2.3$ | $86.5 \pm 1.2$ | $59.2 \pm 0.4$ | $47.1 \pm 0.3$ | $20.2 \pm 0.9$ | $44.7 \pm 0.6$ |
| TopoFilter [42] | $90.4 \pm 0.2$ | $86.8 \pm 0.3$ | $46.8 \pm 1.0$ | $87.5 \pm 0.4$ | $66.9 \pm 0.4$ | $53.4 \pm 1.8$ | $18.3 \pm 1.7$ | $56.6 \pm 0.5$ |
| CRUST [32] | $89.4 \pm 0.2$ | $87.0 \pm 0.1$ | $64.8 \pm 1.5$ | $82.4 \pm 0.0$ | $69.3 \pm 0.2$ | $62.3 \pm 0.2$ | $21.7 \pm 0.7$ | $56.1 \pm 0.5$ |
| FINE | $\mathbf{91.0 \pm 0.1}$ | $\mathbf{87.3 \pm 0.2}$ | $69.4 \pm 1.1$ | $89.5 \pm 0.1$ | $70.3 \pm 0.2$ | $64.2 \pm 0.5$ | $25.6 \pm 1.2$ | $61.7 \pm 1.0$ |
| F-Coteaching | $\mathbf{92.0 \pm 0.1}$ | $\mathbf{87.5 \pm 0.1}$ | $\mathbf{74.2 \pm 0.8}$ | $\mathbf{90.5 \pm 0.2}$ | $\mathbf{71.1 \pm 0.2}$ | $\mathbf{64.7 \pm 0.3}$ | $\mathbf{31.6 \pm 1.0}$ | $\mathbf{64.8 \pm 0.7}$ |

Table 1 summarizes the performances of different sample selection approaches on various noise distribution and datasets. We observe that our FINE method consistently outperforms the competitive methods over the various noise rates. Our FINE methods can filter the clean instances without losing essential information, leading to training the robust network.

To go further, we improve the performance of Co-teaching [14] by substituting its sample selection state with our FINE algorithm. To combine FINE and the Co-teaching family, unlike the original methods that utilize the small loss instances to train with clean labels, we train one model with extracted samples by conducting FINE on another model. The results of the experiments are shown in the eighth and ninth rows of Table 1.

### 4.2.2 SSL-Based Approaches

SSL approaches [25, 10, 41] divide the training data into clean instances as labeled instances and noisy instances as unlabeled instances and use both the labeled and unlabeled samples to train the networks in SSL. Recently, methods belonging to this category have shown the best performance among the various *LNL* methods, and these methods can train robust networks for even extremely high noise rates. We compare the performances of the existing semi-supervised approaches and that in which the sample selection state of DivideMix [25] is substituted with our FINE algorithm (i.e., F-DivideMix). The results of the experiments are shown in Table 2. We achieve consistently higher performance than DivideMix by utilizing FINE instead of its loss-based filtering method and show comparable performance to the state-of-the-art SSL methods such as DST [41] and LongReMix [10]. Interestingly, as Figure 6 shows, clean and noisy data are well classified in F-DivideMix under extreme noise cases.

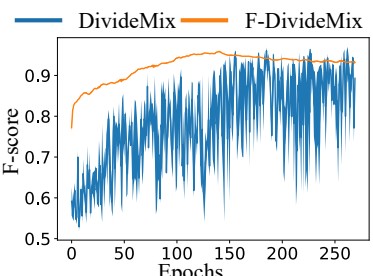

Figure 6: Comparisons of F-scores on CIFAR-10 under symmetric 90% noise. Blue line indicates the error bar of two networks' F-score used in Dividemix [25], and Orange line indicates those replaced by our FINE detector.

### 4.2.3 Collaboration with Noise-Robust Loss Functions

The goal of the noise-robust loss function is to achieve a small risk for unseen clean data even when noisy labels exist in the training data. There have been few collaboration studies of the noise-robust loss function methodology and dynamic sample selection. Most studies have selected clean and noisy data based on cross-entropy loss.

Here, we state the collaboration effects of FINE with various noise-robust loss functions: generalized cross entropy (GCE) [51], symmetric cross entropy (SCE) [40], and early-learning regularization (ELR) [29]. Figure 7 shows that FINE facilitates generalization in the application of noise-robust

Table 2: Comparison of test accuracies (%) for FINE collaborating with DivideMix and existing semi-supervised approaches on CIFAR-10 and CIFAR-100 under different noisy types and fractions. The results for all comparison methods are taken from their original works.

| Dataset | | CIFAR-10 | | | | | CIFAR-100 | | | |
|---|---|---|---|---|---|---|---|---|---|---|
| Noisy Type | | Sym | | | | Asym | Sym | | | |
| Noise Ratio | | 20 | 50 | 80 | 90 | 40 | 20 | 50 | 80 | 90 |
| DivideMix [25] | Best | 96.1 | 94.6 | 93.2 | 76.0 | 93.4 | 77.3 | 74.6 | 60.2 | 31.5 |
| | Last | 95.7 | 94.4 | 92.9 | 75.4 | 92.1 | 76.9 | 74.2 | 59.6 | 31.0 |
| DST [41] | Best | 96.1 | 95.2 | 92.9 | - | 94.3 | 78.0 | 74.3 | 57.8 | - |
| | Last | 95.9 | 94.7 | 92.6 | - | 92.3 | 77.4 | 73.6 | 55.3 | - |
| LongReMix [10] | Best | 96.2 | 95.0 | 93.9 | 82.0 | 94.7 | 77.8 | 75.6 | 62.9 | 33.8 |
| | Last | 96.0 | 94.7 | 93.4 | 81.3 | 94.3 | 77.5 | 75.1 | 62.3 | 33.2 |
| F-DivideMix | Best | 96.1 | 94.9 | 93.5 | **90.5** | 93.8 | **79.1** | 74.6 | 61.0 | **34.3** |
| | Last | 96.0 | 94.5 | 93.2 | **89.6** | 92.8 | **78.8** | 74.3 | 60.1 | **31.2** |

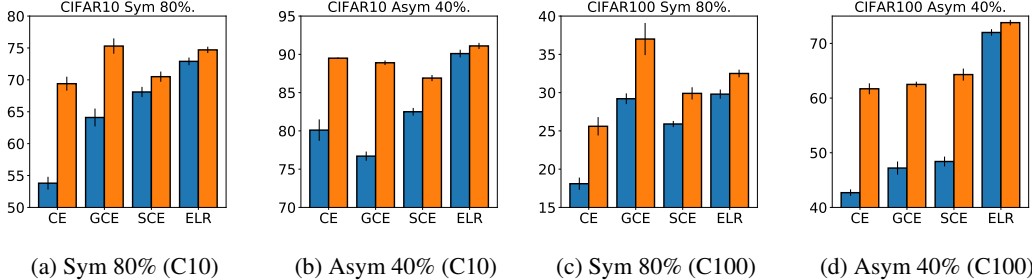

(a) Sym 80% (C10)    (b) Asym 40% (C10)    (c) Sym 80% (C100)    (d) Asym 40% (C100)

Figure 7: Test accuracies (%) on CIFAR-10 and CIFAR-100 under different noisy types and fractions for noise-robust loss approaches. Note that the blue and orange bars are results for without and with FINE, respectively. The average accuracies and standard deviations over three trials are reported.

loss functions on severe noise rate settings. The detailed results are reported in the Appendix. Unlike other methods, it is still theoretically supported because FINE extracts clean data with a robust classifier using representation.

### 4.3 Experiments on Real-World Dataset

As Table 3 shows, FINE and F-DivideMix work fairly well on the Clothing 1M dataset compared to other approaches when we reproduce the experimental results under the same settings.

Table 3: Test accuracy on Clothing1M dataset

| Method | Standard | GCE [51] | SCE [40] | ELR [29] | DivideMix [25] | CORES$^2$ [8] | FINE | F-DivideMix |
|---|---|---|---|---|---|---|---|---|
| Accuracy | 68.94 | 69.75 | 71.02 | 72.87 | 74.30 | 73.24 | 72.91 | **74.37** |

## 5 Conclusion

This paper introduces FINE for detecting label noise by designing a robust noise detector. Our main idea is utilizing the principal components of latent representations made by eigen decomposition. Most existing detection methods are dependent on the loss values, while such losses may be biased by corrupted classifier [24, 31]. Our methodology alleviates this issue by extracting key information from representations without using explicit knowledge of the noise rates. We show that the FINE detector has an excellent ability to detect noisy labels in theoretical and experimental results. We propose three applications of the FINE detector: sample-selection approach, SSL approach, and collaboration with noise-robust loss functions. FINE yields strong results on standard benchmarks and a real-world dataset for various *LNL* approaches.

We believe that our work opens the door to detecting samples having noisy labels with explainable results. It is a non-trivial task and of social significance, and thus, our work will have a substantial social impact on DL practitioners because it avoids the a labor-intensive job of checking data label quality. As future work, we hope that our work will trigger interest in the design of new label-

noise detectors and bring a fresh perspective for other data-resampling approaches (e.g., anomaly detection and novelty detection). The development of robustness against label noise even leads to an improvement in the performance of network trained with data collected through web crawling. We believe that our contribution will lower the barriers to entry for developing robust models for DL practitioners and greatly impact the internet industry. On the other hand, we are concerned that it can be exploited to train robust models using data collected illegally and indiscriminately on the dark web (e.g., web crawling), and thus it may raise privacy concerns (e.g., copyright).

## Acknowledgments and Disclosure of Funding

This work was supported by Institute of Information & communications Technology Planning & Evaluation (IITP) grant funded by the Korea government (MSIT) [No.2019-0-00075, Artificial Intelligence Graduate School Program (KAIST)] and [No. 2021-0-00907, Development of Adaptive and Lightweight Edge-Collaborative Analysis Technology for Enabling Proactively Immediate Response and Rapid Learning]. We thank Seongyoon Kim for discussing about the concept of perturbation.

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
