# A    Theoretical Guarantees for FINE Algorithm

This section provides the detailed proof for Theorem 1 and the lower bounds of the precision and recall. We derive such theorems with the concentration inequalities in probabilistic theory.

## A.1    Preliminaries

**Spectral Norm.**    In this section, we frequently use the spectral norm. For any matrix $\mathbf{A} \in \mathbb{R}^{m \times n}$, the spectral norm are defined as follows:

$$\|\mathbf{A}\|_2 = \sup_{x \in \mathbb{R}^n : \|x\| = 1} \|\mathbf{A}x\|,$$

where $a_{ij}$ is the $(i,j)$ element of $\mathbf{A}$.

**Singular Value Decomposition (SVD).**    Let $\mathbf{A} \in \mathbb{R}^{m \times n}$. There exist orthogonal matrices that satisfy the following:

$$U = [\boldsymbol{u}_1, \boldsymbol{u}_2, \cdots, \boldsymbol{u}_m] \in \mathbb{R}^{m \times m} \quad \text{and} \quad V = [\boldsymbol{v}_1, \boldsymbol{v}_2, \cdots, \boldsymbol{v}_n] \in \mathbb{R}^{n \times n}$$
$$\text{such that} \quad \mathbf{U}^\top \mathbf{A} \mathbf{V} = diag[\sigma_1, \sigma_2, \cdots, \sigma_{min\{m,n\}}] \tag{3}$$

where $\sigma_1 \geq \sigma_2 \geq \cdots \geq \sigma_{min\{m,n\}}$ which are called singular values and $diag[\cdot]$ is a diagonal matrix whose diagonal consists of the vector in the bracket $[\cdot]$. (Note that $\mathbf{U}\mathbf{U}^\top = \mathbf{U}^\top \mathbf{U} = \mathbf{I}$ when $\mathbf{U}$ is an orthogonal matrix).

## A.2    Proof of Theorem 1

We deal with some require lemmas which are used for the proof of Theorem 1.

**Lemma 1.** *Let* $\mathbf{V}$ *and* $\mathbf{W}$ *be orthogonal matrices and* $\mathbf{V} = [\mathbf{V}_1, \mathbf{V}_2]$ *and* $\mathbf{W} = [\mathbf{W}_1, \mathbf{W}_2]$ *with* $\mathbf{V}_1, \mathbf{W}_1 \in \mathbb{R}^{N \times N}$. *Then, we have*

$$\left\|\mathbf{V}_1\mathbf{V}_1^\top - \mathbf{W}_1\mathbf{V}_1^\top\right\|_2 = \left\|\mathbf{V}_1^\top\mathbf{W}_2\right\|_2 = \left\|\mathbf{V}_2^\top\mathbf{W}_1\right\|_2.$$

*Proof.* From the orthogonal invariance property,

$$\left\|\mathbf{V}_1\mathbf{V}_1^\top - \mathbf{W}_1\mathbf{W}_1^\top\right\|_2 = \left\|\mathbf{V}^\top(\mathbf{V}_1\mathbf{V}_1^\top - \mathbf{W}_1\mathbf{W}_1^\top)\mathbf{W}\right\|_2$$
$$= \left\|\begin{bmatrix} 0 & \mathbf{V}_1^\top\mathbf{W}_2 \\ -\mathbf{V}_2^\top\mathbf{W}_1 & 0 \end{bmatrix}\right\|_2$$
$$= \max\{\left\|\mathbf{V}_1^\top\mathbf{W}_2\right\|_2, \left\|\mathbf{V}_2^\top\mathbf{W}_1\right\|_2\},$$

where the last line can be obtained from $\|\mathbf{A}\|_2^2 = \max_{\mathbf{x} \in \mathbb{R}^N : \|\mathbf{x}\|_2 = 1} \|\mathbf{A}\mathbf{x}\|_2^2$.

Thus, to conclude this proof, it suffices to show that $\left\|\mathbf{V}_1^\top\mathbf{W}_2\right\|_2 = \left\|\mathbf{V}_2^\top\mathbf{W}_1\right\|_2$.

Since $\mathbf{W}_1\mathbf{W}_1^\top + \mathbf{W}_2\mathbf{W}_2^\top = \mathbf{I}$,

$$\left\|\mathbf{V}_1^\top\mathbf{W}_2\right\|_2^2 = \max_{\mathbf{x} \in \mathbb{R}^K : \|\mathbf{x}\|_2 = 1} \mathbf{x}^\top\mathbf{V}_1^\top\mathbf{W}_2\mathbf{W}_2^\top\mathbf{V}_1\mathbf{x}$$
$$= \max_{\mathbf{x} \in \mathbb{R}^K : \|\mathbf{x}\|_2 = 1} \mathbf{V}^\top\mathbf{V}_1^\top(\mathbf{I} - \mathbf{W}_1\mathbf{W}_1^\top)\mathbf{V}_1\mathbf{x}$$
$$= \max_{\mathbf{x} \in \mathbb{R}^K : \|\mathbf{x}\|_2 = 1} 1 - \mathbf{x}^\top\mathbf{V}_1^\top\mathbf{W}_1\mathbf{W}_1^\top\mathbf{V}_1\mathbf{x}$$
$$= 1 - \max_{\mathbf{x} \in \mathbb{R}^K : \|\mathbf{x}\|_2 = 1} \mathbf{x}^\top\mathbf{V}_1^\top\mathbf{W}_1\mathbf{W}_1^\top\mathbf{V}_1\mathbf{x}$$
$$= 1 - \lambda_k(\mathbf{V}_1^\top\mathbf{W}_1\mathbf{W}_1^\top\mathbf{V}_1)$$
$$= 1 - \sigma_k(\mathbf{V}_1^\top\mathbf{W}_1)^2,$$

where $\sigma_k(\mathbf{V}_1^\top \mathbf{W}_1)$ is the $k$-th singular value of $\mathbf{V}_1^\top \mathbf{W}_1$. Analogously, we can show that

$$\left\|\mathbf{W}_1^\top \mathbf{V}_2\right\|_2^2 = 1 - \sigma_k(\mathbf{V}_1^\top \mathbf{W}_1)^2.$$

Thus, we have

$$\left\|\mathbf{V}_1\mathbf{V}_1^\top - \mathbf{W}_1\mathbf{W}_1^\top\right\|_2 = \left\|\mathbf{V}_1^\top \mathbf{W}_2\right\|_2 = \left\|\mathbf{V}_2^\top \mathbf{W}_1\right\|_2 = \sqrt{1 - \sigma_k(\mathbf{V}_1^\top \mathbf{W}_1)^2}.$$

$\square$

**Lemma 2.** *(Weyl's Theorem) For any real matrices* $\mathbf{A}, \mathbf{B} \in \mathbb{R}^{m \times n}$,

$$\sigma_i(\mathbf{A} + \mathbf{B}) \le \sigma_i(\mathbf{A}) + \sigma_1(\mathbf{B}).$$

*Proof.* From the definition of the SVD, for any given matrix $\mathbf{X} \in \mathbb{R}^{m \times n}$.

$$\begin{aligned}
\sigma_i(\mathbf{X}) &= \sup_{\mathbf{V}:dim(\mathbf{V})=i} \inf_{\mathbf{v} \in \mathbf{V}:\|\mathbf{v}\|_2=1} \left\|\mathbf{v}^\top(\mathbf{A} + \mathbf{B})\right\|_2 \\
&\le \sup_{\mathbf{V}:dim(\mathbf{V})=i} \inf_{\mathbf{v} \in \mathbf{V}:\|\mathbf{v}\|_2=1} \left\|\mathbf{v}^\top\mathbf{A}\right\|_2 + \left\|\mathbf{v}^\top\mathbf{B}\right\|_2 \\
&\le \sup_{\mathbf{V}:dim(\mathbf{V})=i} \inf_{\mathbf{v} \in \mathbf{V}:\|\mathbf{v}\|_2=1} \left\|\mathbf{v}^\top\mathbf{A}\right\|_2 + \|\mathbf{B}\|_2 \\
&\le \sigma_i(\mathbf{A}) + \sigma_1(\mathbf{B}).
\end{aligned}$$

$\square$

**Lemma 3.** *(David-Kahan sin Theorem) For given symmetric matrices* $\mathbf{A}, \mathbf{B} \in \mathbb{R}^{n \times n}$, *let* $\mathbf{A} = \mathbf{U}\boldsymbol{\Lambda}\mathbf{U}^\top$ *and* $\mathbf{A} + \mathbf{B} = \bar{\mathbf{U}}\bar{\boldsymbol{\Lambda}}\bar{\mathbf{U}}^\top$ *be eigenvalue decomposition of* $\mathbf{A}$ *and* $\mathbf{A} + \mathbf{B}$. *Then,*

$$\left\|\mathbf{U}_{1:k}(\mathbf{U}_{1:k})^\top - \bar{\mathbf{U}}_{1:k}(\bar{\mathbf{U}}_{1:k})^\top\right\|_2 \le \frac{\|\mathbf{B}\|_2}{\lambda_k(\mathbf{A}) - \lambda_{k+1}(\mathbf{A}) - \|\mathbf{B}\|_2},$$

*where* $\mathbf{U}_{1:k}$ *and* $\bar{\mathbf{U}}_{1:k}$ *denote the first k columns of* $\mathbf{U}$ *and* $\bar{\mathbf{U}}$, *respectively.*

*Proof.* Assume that $\mathbf{A}$ and $\mathbf{A} + \mathbf{B}$ have non-negative eigenvalues. If not, there exists a large enough constant $c$ to make $\tilde{\mathbf{A}} + cI$ so that $\tilde{\mathbf{A}}$ and $\tilde{\mathbf{A}} + \mathbf{B}$ become positive semi-definite matrices. Note that $\mathbf{A}$ (resp. $\mathbf{A} + \mathbf{B}$) and $\tilde{\mathbf{A}}$ (resp. $\tilde{\mathbf{A}} + \mathbf{B}$) share the same eigenvectors and eigenvalue gaps $\lambda_i(\mathbf{A}) - \lambda_{i+1}(\mathbf{A})$.

From the *Lemma* 2, we have

$$\lambda_i(\mathbf{A}) - \|\mathbf{B}\|_2 \le \lambda_i(\mathbf{A} + \mathbf{B}) \le \lambda_i(\mathbf{A}) + \|\mathbf{B}\|_2. \tag{4}$$

Thus,

$$\begin{aligned}
(\lambda_{k+1}(\mathbf{A}) + \|\mathbf{B}\|_2)\|(\bar{\mathbf{U}}_{k+1:n})^\top \mathbf{U}_{k+1:n}\|_2 &\ge \|(\bar{\mathbf{U}}_{k+1:n})^\top(\mathbf{A} + \mathbf{B})\mathbf{U}_{1:k}\|_2 & (5) \\
&\ge \|(\bar{\mathbf{U}}_{k+1:n})^\top \mathbf{A}\mathbf{U}_{1:k}\|_2 - \|\mathbf{B}\|_2 & (6) \\
&\ge \lambda_k(\mathbf{A})\|(\bar{\mathbf{U}}_{k+1:n})^\top \mathbf{U}_{1:k}\|_2 - \|\mathbf{B}\|_2 & (7)
\end{aligned}$$

From (7), we have

$$\|\mathbf{U}_{1:k}(\mathbf{U}_{1:k})^\top - \bar{\mathbf{U}}_{1:k}(\bar{\mathbf{U}}_{1:k})^\top\|_2 \le \frac{\|\mathbf{B}\|_2}{\lambda_k(\mathbf{A}) - \lambda_{k+1}(\mathbf{A}) - \|\mathbf{B}\|_2}.$$

*Proof of (5)* : Since the columns of $\bar{\mathbf{U}}_{k+1:n}$ are singular vectors of $\mathbf{A} + \mathbf{B}$,

$$(\bar{\mathbf{U}}_{k+1:n})^\top (\mathbf{A} + \mathbf{B})\mathbf{U}_{1:k} = \bar{\boldsymbol{\Lambda}}_{k+1:n}(\bar{\mathbf{U}}_{k+1:n})^\top \mathbf{U}_{1:k}.$$

Therefore,

$$\left\|(\bar{\mathbf{U}}_{k+1:n})^\top (\mathbf{A} + \mathbf{B})\mathbf{U}_{1:k}\right\|_2 \le \left\|\bar{\boldsymbol{\Lambda}}_{k+1:n}\right\|_2 \left\|(\bar{\mathbf{U}}_{k+1:n})^\top \mathbf{U}_{1:k}\right\|_2 = \lambda_{k+1}(\mathbf{A}+\mathbf{B})\left\|(\bar{\mathbf{U}}_{k+1:n})^\top \mathbf{U}_{1:k}\right\|_2$$

From (4), we have $\lambda_{k+1}(\mathbf{A} + \mathbf{B}) \le \lambda_{k+1}(\mathbf{A}) + \|\mathbf{B}\|_2$

*Proof of (6)* : From the triangle inequality,

$$\begin{aligned}
\left\|(\bar{\mathbf{U}}_{k+1:n})^\top \mathbf{A}\mathbf{U}_{1:k}\right\|_2 &= \left\|(\bar{\mathbf{U}}_{k+1:n})^\top (\mathbf{A} + \mathbf{B})\mathbf{U}_{1:k} + (-\bar{\mathbf{U}}_{k+1:n})^\top \mathbf{B}\mathbf{U}_{1:k}\right\|_2 \\
&\le \left\|\bar{\mathbf{U}}_{k+1:n})^\top (\mathbf{A} + \mathbf{B})\mathbf{U}_{1:k}\right\|_2 + \left\|(-\bar{\mathbf{U}}_{k+1:n})^\top \mathbf{B}\mathbf{U}_{1:k}\right\|_2 .
\end{aligned}$$

We have

$$\left\|(-\bar{\mathbf{U}}_{k+1:n})^\top \mathbf{B}\mathbf{U}_{1:k}\right\|_2 \le \left\|(-\bar{\mathbf{U}}_{k+1:n})^\top\right\|_2 \|\mathbf{B}\|_2 \|\mathbf{U}_{1:k}\|_2 = \|\mathbf{B}\|_2 .$$

*Proof of (7)* : Since the columns of $\mathbf{U}_{1:k}$ are singular vectors of $\mathbf{A}$,

$$(\bar{\mathbf{U}}_{k+1:n})^\top \mathbf{A}\mathbf{U}_{1:k} = (\bar{\mathbf{U}}_{k+1:n})^\top \mathbf{U}_{1:k}\boldsymbol{\Lambda}_{1:k}$$

Therefore,

$$\left\|(\bar{\mathbf{U}}_{k+1:n})^\top \mathbf{A}\mathbf{U}_{1:k}\right\|_2 = \left\|(\bar{\mathbf{U}}_{k+1:n})^\top \mathbf{U}_{1:k}\boldsymbol{\Lambda}_{1:k}\right\|_2 \ge \left\|(\bar{\mathbf{U}}_{k+1:n})^\top \mathbf{U}_{1:k}\right\|_2 \lambda_k(\mathbf{A}).$$

$\square$

**Lemma 4.** *Let $\mathbf{M} \in S^{d \times d}$ and let $N_\epsilon$ be an $\epsilon$-net of $\mathbb{S}^{d-1}$. Then*

$$\|\mathbf{M}\|_2 \le \frac{1}{1 - 2\epsilon} \max_{\boldsymbol{y} \in N_\epsilon} |\boldsymbol{y}^\top \mathbf{M}\boldsymbol{y}|$$

*Proof.* Let $\mathbf{M} \in S^{d \times d}$ and let $N_\epsilon$ be an $\epsilon$-net of $\mathbb{S}^{d-1}$. Furthermore, we define $\boldsymbol{y} \in N_\epsilon$ satisfy $\|\boldsymbol{x} - \boldsymbol{y}\|_2 \le \epsilon$. Then,

$$\begin{aligned}
|\boldsymbol{x}^{\mathbf{M}}\boldsymbol{x} - \boldsymbol{y}^\top \mathbf{M}\boldsymbol{y}| &= |\boldsymbol{x}^\top \mathbf{M}(\boldsymbol{x} - \boldsymbol{y}) + \boldsymbol{y}^\top \mathbf{M}(\boldsymbol{x} - \boldsymbol{y})| \\
&\le |\boldsymbol{x}^\top \mathbf{M}(\boldsymbol{x} - \boldsymbol{y})| + |\boldsymbol{y}^\top \mathbf{M}(\boldsymbol{x} - \boldsymbol{y})|
\end{aligned} \tag{8}$$

Looking at $|\boldsymbol{x}^\top \mathbf{M}(\boldsymbol{x} - \boldsymbol{y})|$ we have

$$\begin{aligned}
|\boldsymbol{x}^\top \mathbf{M}(\boldsymbol{x} - \boldsymbol{y})| &\le \|\mathbf{M}(\boldsymbol{x} - \boldsymbol{y})\|_2 \|\boldsymbol{x}\|_2 \\
&\le \|\mathbf{M}\|_2 \|\boldsymbol{x} - \boldsymbol{y}\|_2 \|\boldsymbol{x}\|_2 \\
&\le \|\mathbf{M}\|_2 \epsilon
\end{aligned} \tag{9}$$

Applying the same argument to $\boldsymbol{y}^\top \mathbf{M}(\boldsymbol{x} - \boldsymbol{y})|$ gives us $|\boldsymbol{x}^{\mathbf{M}}\boldsymbol{x} - \boldsymbol{y}^\top \mathbf{M}\boldsymbol{y}| \le 2\epsilon \|\mathbf{M}\|_2$. To complete the proof, we see that $\|\mathbf{M}\|_2 = \max_{\boldsymbol{x} \in \mathbb{S}^{d-1}} \boldsymbol{x}^\top \mathbf{M}\boldsymbol{x} \le 2\epsilon \|\mathbf{M}\|_2 + \max_{\boldsymbol{y} \in N_\epsilon} \boldsymbol{y}^\top \mathbf{M}\boldsymbol{y}$. Rearranging the equation gives $\|\mathbf{M}\|_2 \le \frac{1}{1-2\epsilon} \max_{\boldsymbol{y} \in N_\epsilon} \boldsymbol{y}^\top \mathbf{M}\boldsymbol{y}$ as desired. $\square$

**Lemma 5.** *Let $x_1, \ldots, x_n$ be an i.i.d sequence of $\sigma$ sub-gaussian random vectors such that $\mathbb{V}[x_1] = \Sigma$ and let $\hat{\Sigma}_n := \frac{1}{N} \sum_{i=1}^n x_i x_i^\top$ be the empirical gram matrix. Then, there exists a universal constant $C > 0$ such that, for $\delta \in (0, 1)$, with probability at least $1 - \delta$*

$$\frac{\left\| \hat{\Sigma}_n - \Sigma \right\|_2}{\sigma^2} \leq C \max\{ \sqrt{\frac{d + \log(2/\delta)}{n}}, \frac{d + \log(2/\delta)}{n} \}$$

*Proof.* Applying Lemma 4 on $\hat{\Sigma}_n - \Sigma$ with $\epsilon = 1/4$ we have

$$\left\| \hat{\Sigma}_n - \Sigma \right\|_2 \leq 2 \max_{v \in N_{1/4}} |v^\top \hat{\Sigma}_n - \Sigma v|$$

Additionally, we know that $N_{1/4} \leq 9^d$. From here, we can apply standard concentration tools as follows:

$$
\begin{aligned}
\mathbb{P}(\left\| \hat{\Sigma}_n - \Sigma \right\|_2 \geq t) &\leq \mathbb{P}(\max_{v \in N_{1/4}} |v^\top (\hat{\Sigma}_n - \Sigma) v| \geq t/2) \\
&\leq |N_{1/4}| \mathbb{P}(|v_i^\top (\hat{\Sigma}_n - \Sigma) v_i| \geq t/2)
\end{aligned}
\tag{10}
$$

We rewrite $v_i^\top (\hat{\Sigma}_n - \Sigma) v_i$ as follows:

$$
\begin{aligned}
v_i^\top (\hat{\Sigma}_n - \Sigma) v_i &= \frac{1}{n} \sum_{i=1}^n (v_i^\top x_j)^2 - \mathbb{E}\left[(v_i^\top x_j)^2\right] \\
&= \frac{1}{n} \sum_{i=1}^n z_j - \mathbb{E}[z_j]
\end{aligned}
$$

where $z_j$'s are independent and by assumption $v_i^\top x_j \in SG(\sigma^2)$ so that $z_j - \mathbb{E}[z_j] \in SE((16\sigma^2)^2, 16\sigma^2)$. Applying the sub-exponential tail bound gives us

$$\mathbb{P}(v_i^\top (\hat{\Sigma}_n - \Sigma) v_i| \geq t/2) \leq 2 \exp\left[ -\frac{n}{2} \min\left\{ \frac{n}{(32\sigma^2)^2}, \frac{n}{32\sigma^2} \right\} \right]$$

so that

$$\mathbb{P}(\left\| \hat{\Sigma}_n - \Sigma \right\|_2 \geq t) \leq 2 \cdot 9^d 2 \exp\left[ -\frac{n}{2} \min\left\{ \frac{n}{(32\sigma^2)^2}, \frac{n}{32\sigma^2} \right\} \right]$$

Inverting the bound gives the desired result. $\square$

### A.2.1 Proof of Theorem 1

*Proof.* Let $v_\perp$ be the unit vector, which is orthogonal to $v$. Then, $w$ can be expressed by $v_\perp$ and $v$ (i.e. $w = \cos\theta \cdot v + \sin\theta \cdot v_\perp$ with $-\pi/2 \leq \theta \leq \pi/2$). Since $vv^\top + v_\perp v_\perp^\top = I$, we have $w = vv^\top w + v_\perp v_\perp^\top w$.

Then, we have

$$
\begin{aligned}
ww^\top &= (vv^\top w + v_\perp v_\perp^\top w)(vv^\top w + v_\perp w_\perp^\top w)^\top \\
&= vv^\top ww^\top vv^\top + v_\perp v_\perp^\top ww^\top vv^\top + vv^\top ww^\top v_\perp v_\perp^\top + v_\perp v_\perp^\top ww^\top v_\perp v_\perp^\top
\end{aligned}
\tag{11}
$$

Let $A$ and $B$ be the projection matrices for clean instances and whole instances for using the *David-Kahan sin Theorem* as followings:

$$\mathbf{A} = \mathbb{E}\left[\sum_{i=1}^{N_+}(\boldsymbol{v}+\epsilon_i)(\boldsymbol{v}+\epsilon_i)^\top\right] + \boldsymbol{v}_\perp \boldsymbol{v}_\perp^\top \boldsymbol{w}\boldsymbol{w}^\top \boldsymbol{v}_\perp \boldsymbol{v}_\perp^\top + \sigma^2 \mathbf{I} \tag{12}$$

$$\mathbf{A} + \mathbf{B} = \sum_{i=1}^{N_+}(\boldsymbol{v}+\epsilon_i)(\boldsymbol{v}+\epsilon_i)^\top + \sum_{j=1}^{N_-}(\boldsymbol{w}+\epsilon_j)(\boldsymbol{w}+\epsilon_j)^\top \tag{13}$$

The difference between first eigenvalue and second eigenvalue of gram matrix $\mathbf{A}$ is equal to

$$\lambda_1(\mathbf{A}) - \lambda_2(\mathbf{A}) = N_+ - N_- \sin^2\theta \geq N_+ - N_- \sin\theta \tag{14}$$

By triangular inequality, we have

$$\|\mathbf{B}\|_2 \leq \left\|\sum_{i=1}^{N_+}(\boldsymbol{v}+\epsilon_i)(\boldsymbol{v}+\epsilon_i)^\top - \boldsymbol{v}\boldsymbol{v}^\top - \sigma^2\mathbf{I}\right\|_2 + \left\|\sum_{i=1}^{N_-}(\boldsymbol{w}+\epsilon_i)(\boldsymbol{w}+\epsilon_i)^\top - \boldsymbol{w}\boldsymbol{w}^\top - \sigma^2\mathbf{I}\right\|_2$$
$$+ \left\|\sum_{j=1}^{N_-}\boldsymbol{w}\boldsymbol{w}^\top - \boldsymbol{v}_\perp^\top \boldsymbol{w}\boldsymbol{w}^\top \boldsymbol{v}_\perp \boldsymbol{v}_\perp^\top\right\|_2 \tag{15}$$

For the first and the second terms of RHS in Eq. (15), using Lemma 5, with probability at least $1 - \delta/2$, we can derive each term as followings:

$$\left\|\sum_{i=1}^{N_+}(\boldsymbol{v}+\epsilon_i)(\boldsymbol{v}+\epsilon_i)^\top - \boldsymbol{v}\boldsymbol{v}^\top - \sigma^2\mathbf{I}\right\|_2 \leq N_+ C\sigma^2 \max\left\{\sqrt{\frac{d+\log(4/\delta)}{N_+}}, \frac{d+\log(4/\delta)}{N_+}\right\},$$

$$\left\|\sum_{j=1}^{N_-}(\boldsymbol{w}+\epsilon_j)(\boldsymbol{w}+\epsilon_j)^\top - \boldsymbol{w}\boldsymbol{w}^\top - \sigma^2\mathbf{I}\right\|_2 \leq N_- C\sigma^2 \max\left\{\sqrt{\frac{d+\log(4/\delta)}{N_-}}, \frac{d+\log(4/\delta)}{N_-}\right\}$$

As $N_+, N_- \to \infty$, with probability at least $1 - \delta$

$$\frac{1}{N_+}\left[\left\|\sum_{i=1}^{N_+}(\boldsymbol{v}+\epsilon_i)(\boldsymbol{v}+\epsilon_i)^\top - \boldsymbol{v}\boldsymbol{v}^\top - \sigma^2\mathbf{I}\right\|_2 + \left\|\sum_{j=1}^{N_-}(\boldsymbol{w}+\epsilon_j)(\boldsymbol{w}+\epsilon_j)^\top - \boldsymbol{w}\boldsymbol{w}^\top - \sigma^2\mathbf{I}\right\|_2\right]$$
$$\leq C\sigma^2\left[\sqrt{\frac{d+\log(4/\delta)}{N_+}} + \frac{N_-}{N_+}\sqrt{\frac{d+\log(4/\delta)}{N_-}}\right] = \mathcal{O}\left(\sigma^2\sqrt{\frac{d+\log(4/\delta)}{N_+}} + \sigma^2\tau\sqrt{\frac{d+\log(4/\delta)}{N_-}}\right) \tag{16}$$

For the third term of RHS in Eq. (15), we have

$$\left\|\sum_{j=1}^{N_-}\boldsymbol{w}\boldsymbol{w}^\top - \boldsymbol{v}_\perp^\top \boldsymbol{w}\boldsymbol{w}^\top \boldsymbol{v}_\perp \boldsymbol{v}_\perp^\top\right\|_2 = N_- \cdot \left\|\boldsymbol{v}\boldsymbol{v}^\top \boldsymbol{w}\boldsymbol{w}^\top \boldsymbol{v}\boldsymbol{v}^\top + \boldsymbol{v}_\perp \boldsymbol{v}_\perp^\top \boldsymbol{w}\boldsymbol{w}^\top \boldsymbol{v}\boldsymbol{v}^\top + \boldsymbol{v}\boldsymbol{v}^\top \boldsymbol{w}\boldsymbol{w}^\top \boldsymbol{v}_\perp \boldsymbol{v}_\perp^\top\right\|_2$$
$$\leq N_- \cdot 3\cos\theta \tag{17}$$

Hence, by using Eq. (14), Eq. (16), and Eq. (17) for Lemma 3, when $\tau$ is sufficiently small, we have

$$\left\| \boldsymbol{u}\boldsymbol{u}^\top - \boldsymbol{v}\boldsymbol{v}^\top \right\|_2 \leq \frac{3\tau\cos\theta + \mathcal{O}(\sigma^2\sqrt{\frac{d+\log(4/\delta)}{N_+}})}{1 - \tau(\sin\theta + 3\cos\theta) - \mathcal{O}(\sigma^2\sqrt{\frac{d+\log(4/\delta)}{N_+}})}. \tag{18}$$

$\square$

### A.3 Additional Theorem

After projecting the features on the principal component of FINE detector, we aim to guarantee the lower bounds for the precision and recall of such values with high probability. Since the feature distribution comprises two Gaussian distributions, the projected distribution is also a mixture of two Gaussian distributions. Here, by LDA assumptions, its decision boundary with $\zeta = 0.5$ is the same as the average of mean of two clusters. In this situation, we provide the lower bounds for the precision and recall of our FINE detector in Theorem 2.

**Theorem 2.** *Let $\Phi$ be the cumulative distribution function (CDF) of $\mathcal{N}(0,1)$. Additionally, we define the $z_i$ as linear projection of $\mathbf{x}_i$ on arbitrary vector. For the decision boundary $b = \frac{1}{2}(\frac{\sum_{i=1}^N \mathbb{1}_{\{y_i=1\}} z_i}{N_+} + \frac{\sum_{i=1}^N \mathbb{1}_{\{y_i=-1\}} z_i}{N_-})$, with probability 1-$\delta$, the lower bounds for the precision and recall can be derived as Eq. (21) and Eq. (22).*

*Proof.* $N_+$ and $N_-$ are equal to $\sum_{i=1}^N \mathbb{1}_{\{Y_i=1\}}$ and $\sum_{i=1}^N \mathbb{1}_{\{Y_i=-1\}}$, respectively. With definition 2, the mean of the projection values of clean instances is $(\boldsymbol{u}^\top\boldsymbol{v})^2$ and that of noisy instances is $(\boldsymbol{u}^\top\boldsymbol{w})^2$. By the central limit theorem (CLT), we have $\frac{\sum_{i=1}^N \mathbb{1}_{\{Y_i=1\}} Z_i}{N_+} \sim \mathcal{N}((\boldsymbol{u}^\top\boldsymbol{v})^2, \frac{\sigma^2}{N_+})$ and $\frac{\sum_{i=1}^N \mathbb{1}_{\{Y_i=-1\}} Z_i}{N_-} \sim \mathcal{N}((\boldsymbol{u}^\top\boldsymbol{w})^2, \frac{\sigma^2}{N_-})$. Furthermore, we can get $\frac{\sum_{i=1}^N \mathbb{1}_{\{Y_i=1\}} Z_i}{N_+} + \frac{\sum_{i=1}^N \mathbb{1}_{\{Y_i=-1\}} Z_i}{N_-} \sim \mathcal{N}((\boldsymbol{u}^\top\boldsymbol{v})^2 + (\boldsymbol{u}^\top\boldsymbol{w})^2, (\frac{1}{N_+} + \frac{1}{N_-})\sigma^2)$

By the concentration inequality on standard Gaussian distribution, we have

$$\mathbb{P}\left(\left|\frac{\sum_{i=1}^N \mathbb{1}_{\{Y_i=1\}} Z_i}{N_+} + \frac{\sum_{i=1}^N \mathbb{1}_{\{Y_i=-1\}} Z_i}{N_-} - ((\boldsymbol{u}^\top\boldsymbol{v})^2 + (\boldsymbol{u}^\top\boldsymbol{w})^2)\right| > \psi\right) < 2\exp\left(-\frac{\psi^2}{2} \cdot \frac{1}{\frac{\sigma^2}{N_+} + \frac{\sigma^2}{N_-}}\right) \tag{19}$$

Therefore, with probability $1 - \delta$,

$$\frac{(\boldsymbol{u}^\top\boldsymbol{v})^2 + (\boldsymbol{u}^\top\boldsymbol{w})^2}{2} - \mathcal{C}\sqrt{\left(\frac{1}{N_+} + \frac{1}{N_+}\right)\log(2/\delta)} \leq b$$

$$b \leq \frac{(\boldsymbol{u}^\top\boldsymbol{v})^2 + (\boldsymbol{u}^\top\boldsymbol{w})^2}{2} + \mathcal{C}\sqrt{\left(\frac{1}{N_+} + \frac{1}{N_+}\right)\log(2/\delta)} \tag{20}$$

where $\mathcal{C} > 0$ is a constant. Then, by using the Eq. (20), we can derive the lower bound for the recall as follows:

$$
\begin{aligned}
\text{RECALL} = \mathbb{P}(Z > b | Y = +1) &= P(Z > b | Y = +1) \\
&\geq \mathbb{P}(Z > \frac{\mu_+ + \mu_-}{2} + \mathcal{C}\sqrt{\left(\frac{1}{N_+} + \frac{1}{N_+}\right)\log(2/\delta)} | Y = +1) \\
&= \mathbb{P}(\frac{Z - \mu_+}{\sigma} > \frac{-\mu_+ + \mu_-}{2\sigma} + \frac{\mathcal{C}\sqrt{\left(\frac{1}{N_+} + \frac{1}{N_+}\right)\log(2/\delta)}}{\sigma}) \\
&= \mathbb{P}(\mathcal{N}(0,1) > \frac{-\Delta + 2\mathcal{C}\sqrt{\left(\frac{1}{N_+} + \frac{1}{N_+}\right)\log(2/\delta)}}{2\sigma}) \\
&= 1 - \mathbb{P}(\mathcal{N}(0,1) \leq \frac{-\Delta + 2\mathcal{C}\sqrt{\left(\frac{1}{N_+} + \frac{1}{N_+}\right)\log(2/\delta)}}{2\sigma}) \\
&= 1 - \Phi(\frac{-\Delta + 2\mathcal{C}\sqrt{\left(\frac{1}{N_+} + \frac{1}{N_+}\right)\log(2/\delta)}}{2\sigma}) \\
&= \Phi(\frac{\Delta - 2\mathcal{C}\sqrt{\left(\frac{1}{N_+} + \frac{1}{N_+}\right)\log(2/\delta)}}{2\sigma})
\end{aligned}
\tag{21}
$$

Furthermore, we have lower bound for precision as follows:

$$
\begin{aligned}
\text{PRECISION} &= \mathbb{P}(Y = +1 | Z > b) \\
&= \frac{\mathbb{P}(Z > b | Y = +1)P(Y = +1)}{\sum_{i \in \{-1,+1\}} \mathbb{P}(Z > b | Y = i)P(Y = i)} \\
&\geq \frac{\mathbb{P}(Z > \frac{\mu_+ + \mu_-}{2} + \mathcal{C}\sqrt{\left(\frac{1}{N_+} + \frac{1}{N_+}\right)\log(2/\delta)} | Y = +1)P(Y = +1)}{\sum_{i \in \{-1,+1\}} \mathbb{P}(Z > \frac{\mu_+ + \mu_-}{2} - \mathcal{C}\sqrt{\left(\frac{1}{N_+} + \frac{1}{N_+}\right)\log(2/\delta)} | Y = i)P(Y = i)} \\
&= \frac{\mathbb{P}(Z > \frac{\mu_+ + \mu_-}{2} + \mathcal{C}\sqrt{\left(\frac{1}{N_+} + \frac{1}{N_+}\right)\log(2/\delta)} | Y = +1)P(Y = +1)}{\sum_{i \in \{-1,+1\}} \mathbb{P}(Z > \frac{\mu_+ + \mu_-}{2} - \mathcal{C}\sqrt{\left(\frac{1}{N_+} + \frac{1}{N_+}\right)\log(2/\delta)} | Y = i)P(Y = i)} \\
&\geq \frac{1}{1 + \frac{\mathbb{P}(Z > \frac{\mu_+ + \mu_-}{2} + \mathcal{C}\sqrt{\left(\frac{1}{N_+} + \frac{1}{N_+}\right)\log(2/\delta)} | Y = -1)P(Y = -1)}{\mathbb{P}(Z > \frac{\mu_+ + \mu_-}{2} - \mathcal{C}\sqrt{\left(\frac{1}{N_+} + \frac{1}{N_+}\right)\log(2/\delta)} | Y = +1)P(Y = +1)}} \\
&= \frac{1}{1 + \frac{p_- \cdot \Phi(\frac{-\Delta - 2\mathcal{C}\sqrt{\left(\frac{1}{N_+} + \frac{1}{N_+}\right)\log(2/\delta)}}{2\sigma})}{p_+ \cdot \Phi(\frac{\Delta - 2\mathcal{C}\sqrt{\left(\frac{1}{N_+} + \frac{1}{N_+}\right)\log(2/\delta)}}{2\sigma})}}
\end{aligned}
\tag{22}
$$

where $\Delta := \boldsymbol{u}^\top \boldsymbol{v} - \boldsymbol{u}^\top \boldsymbol{w}$, and $p_+$ and $p_-$ are the noise distribution for clean instances and noisy instances, respectively. Additionally, we can find that the difference of mean between two Gaussian

distribution, $\Delta > 0$ is an important factor of computing both lower bounds. As $\Delta$ become larger, we have larger lower bounds for both recall and precision. □

# B   Implementation Details for section 4

In section 4, there are two reporting styles regarding the test accuracy: (1) reporting the accuracy with statistics and (2) reporting the best and last test accuracy. For the first one, we leverage an additional validation set to select the best model [14, 29], and thus the reported accuracy is computed with this selected model. In the next case, we report both the best and last test accuracy without the usage of validation set [25]. We reproduce all experimental results referring to other official repositories [3, 4, 5].

**Dataset.**   In our expereiments, we compare the methods regarding image classification on three benchmark datasets: CIFAR-10, CIFAR-100, and Clothing-1M [44][6]. Because CIFAR-10, CIFAR-100 do not have predefined validation sets, we retain 10% of the training sets to perform validation [29].

**Data Preprocessing**   We use the same settings in [29]. We apply normalization and simple data augmentation techniques (random crop and horizontal flip) on the training sets of all datasets. The size of the random crop is set to 32 for the CIFAR datasets and 224 for Clothing1M referred to previous works [29, 51, 19].

## B.1   Sample-Selection Approaches

As an extension on the experiments in original papers [14, 47, 42, 32], we conduct experiments on various noise settings. We use the same hyperparameter settings written in each paper (algorithm 2). Therefore, we unify the hyperparameter settings. In this experiment, we use ResNet-34 models and reported their accuracy. For using FINE detector, we substitute the Topofilter [42] with FINE. Specifically, we use 40 epochs for warmup stage, and the data selection using FINE detector is performed every 10 epochs for computational efficiency referred to the alternative method [42]. The other settings are the same with them.

## B.2   Semi-Superivsed Approaches

DivideMix [25] solves a noisy classification challenge as semi-supervised approach. It trains two separated networks to avoid confirmation errors. The training pipeleine consists of *co-divide* phase and semi-supervised learning (SSL) phase. Firstly, in *co-divide* phase, two networks divide the whole training set into clean and noisy subset and provide them to each other. In SSL phase, each network utilizes clean and noisy subset as labeled and unlabeled training set, respectively, and do the Mix-Match [4] after processing label adjustment, *co-refinement* and *co-guessing*. It adjusts the labels of given samples with each model's prediction, and this adjustment can be thought as a label smoothing for robust training.

In *co-divide* phase, each network calculates cross-entropy (CE) loss value of each training sample and fits them into Gaussian Mixture Model (GMM) with two components which indicate the distribution of clean and noisy subsets. From this process, each sample has clean probability which means how close the sample is to the 'clean' components of GMM.

We demonstrate that FINE detector may be a substitute for the noisy detector in *co-divide* phase (algorithm 3). In every training epoch in DivideMix, the noisy instances are filtered through our FINE detector. algorithm 3 represents the details about the modified algorithm, written based on Dividemix original paper. All hyper-parameters settings are the same with [25], even for the clean probability threshold $\zeta$.

---

[3]`https://github.com/bhanML/Co-teaching`
[4]`https://github.com/LiJunnan1992/DivideMix`
[5]`https://github.com/shengliu66/ELR`
[6]This dataset is not public, and thus we contact the main owner of this dataset to access this dataset. Related procedures are in `https://github.com/Cysu/noisy_label`.

**Algorithm 2:** *Sample-Selection* with FINE

INPUT : weight parameters of a network $\theta$, $\mathcal{D} = (\mathcal{X}, \mathcal{Y})$: training set, number of classes $K$
OUTPUT : $\theta$
1: $\theta = \text{WarmUp}(\mathcal{X}, \mathcal{Y}, \theta)$
2: **while** $e < MaxEpoch$ **do**
3:     **for** $(\boldsymbol{x}_i, y_i) \in \mathcal{D}$ **do**
4:         $\boldsymbol{z}_i \leftarrow g(\boldsymbol{x}_i)$
5:         Update the gram matrix $\boldsymbol{\Sigma}_{y_i} \leftarrow \boldsymbol{\Sigma}_{y_i} + \boldsymbol{z}_i \boldsymbol{z}_i^\top$
6:     **end for**
    /* Generate the principal component with eigen decomposition */
7:     **for** $k = 1, \ldots, K$ **do**
8:         $\mathbf{U}_k, \boldsymbol{\Lambda}_k \leftarrow$ EIGEN DECOMPOSITION OF $\boldsymbol{\Sigma}_k$
9:         $\mathbf{u}_k \leftarrow$ THE FIRST COLUMN OF $\mathbf{U}_k$
10:    **end for**
    /* Compute the alignment score and get clean subset $\mathcal{C}$ */
11:    **for** $(\boldsymbol{x}_i, y_i) \in \mathcal{C}_{e-1}$ **do**
12:        Compute the FINE score $f_i = \langle \mathbf{u}_{y_i}, \boldsymbol{z}_i \rangle^2$ and $\mathcal{F}_{y_i} \leftarrow \mathcal{F}_{y_i} \cup \{f_i\}$
13:    **end for**
    /* Finding the samples whose clean probability is larger than $\zeta$ */
14:    $\mathcal{C}_e \leftarrow \mathcal{C}_e \cup$ GMM $(\mathcal{F}_k, \zeta)$ for all $k = 1, \ldots, K$
15:    Train network $\theta$ using loss function $\mathcal{L}$ on $\mathcal{C}_e$
16: **end while**

---

**Algorithm 3:** *DivideMix* [25] with FINE

INPUT : $\theta^{(1)}$ and $\theta^{(2)}$: weight parameters of two networks, $\mathcal{D} = (\mathcal{X}, \mathcal{Y})$: training set , $\tau$: clean probability threshold, $M$: number of augmentations, $T$: sharpening temperature, $\lambda_u$: unsupervised loss weight, $\alpha$: Beta distribution parameter for MixMatch, FINE
OUTPUT : $\theta^{(1)}$ and $\theta^{(2)}$
1: $\theta^{(1)}, \theta^{(2)} = \text{WarmUp}(\mathcal{X}, \mathcal{Y}, \theta^{(1)}, \theta^{(2)})$
2: **while** $e < MaxEpoch$ **do**
3:     $\mathcal{C}_e^{(2)}, \mathcal{W}^{(2)} = \text{FINE}(\mathcal{X}, \mathcal{Y}, \theta^{(1)})$        $\triangleright$ $\mathcal{W}^{(2)}$ is a set of the probabilities from $\theta^{(2)}$ model
4:     $\mathcal{C}_e^{(1)}, \mathcal{W}^{(1)} = \text{FINE}(\mathcal{X}, \mathcal{Y}, \theta^{(2)})$        $\triangleright$ $\mathcal{W}^{(1)}$ is a set of the probabilities from $\theta^{(1)}$ model
5:     **for** $k = 1, 2$ **do**
6:         $\mathcal{X}_e^{(k)} = \left\{ (x_i, y_i, w_i) | w_i \geq \tau, (x_i, y_i) \in \mathcal{C}_e^{(k)}, \forall (x_i, y_i, w_i) \in (\mathcal{X}, \mathcal{Y}, \mathcal{W}^{(k)}) \right\}$
7:         $\mathcal{U}_e^{(k)} = \mathcal{D} - \mathcal{X}_e^{(k)}$
8:         **for** $b = 1$ to $B$ **do**
9:             **for** $m = 1$ to $M$ **do**
10:              $\hat{x}_{b,m} = Augment(x_b)$
11:              $\hat{u}_{b,m} = Augment(u_b)$
12:            **end for**
13:            $p_b = \frac{1}{M} \Sigma_m p_{model}(\hat{x}_{b,m}; \theta^{(k)})$
14:            $\bar{y}_b = w_b y_b + (1 - w_b) p_b$
15:            $\hat{y}_b = Sharpen(\bar{y}_b, T)$
16:            $\bar{q}_b = \frac{1}{2M} \Sigma_m (p_{model}(\hat{u}_{b,m}; \theta^{(1)}) + p_{model}(\hat{u}_{b,m}; \theta^{(2)}))$
17:            $q_b = Sharpen(\bar{q}_b, T)$
18:         **end for**
19:         $\hat{\mathcal{X}} = \{(\hat{x}_{b,m}, \hat{y}_b); b \in (1, \ldots, B), m \in (1, \ldots, M)\}$
20:         $\hat{\mathcal{U}} = \{(\hat{u}_{b,m}, \hat{y}_b); b \in (1, \ldots, B), m \in (1, \ldots, M)\}$
21:         $\mathcal{L}_\mathcal{X}, \mathcal{L}_\mathcal{U} = MixMatch(\hat{\mathcal{X}}, \hat{\mathcal{U}})$
22:         $\mathcal{L} = \mathcal{L}_\mathcal{X} + \lambda_u \mathcal{L}_\mathcal{U} + \lambda_r \mathcal{L}_{reg}$
23:         $\theta^{(k)} = SGD(\mathcal{L}, \theta^{(k)})$
24:     **end for**
25: **end while**

## B.3 Collaboration with Noise-Robust Loss Functions

We conduct experiments with CE, GCE, SCE, ELR mentioned in subsubsection 4.2.3. We follow all experiments settings presented in the [29] except for the GCE on CIFAR-100 dataset. We use ResNet-34 models and trained them using a standard Pytorch SGD optimizer with a momentum of 0.9. We set a batch size of 128 for all experiments. We utilize weight decay of 0.001 and set the initial learning rate as 0.02, and reduce it by a factor of 100 after 40 and 80 epochs for CIFAR-10 (total 120 epochs) and after 80 and 120 epochs for CIFAR-100 (total 150 epochs). For noise-robust loss functions, we train the network naively for 50 epochs, and conduct the FINE for every 10 epochs.

## C  More Results

### C.1  Degree of Alignment

We additionally explain the vital role of the first eigenvector compared to other eigenvectors and the mean vector.

**Comparison to other eigenvectors.**   We provide robustness by means of the way the first eigenvector is robust to noisy vectors so that FINE can fix the noisy classifier by using segregated clean data. Unlike the first eigenvector, the other eigenvectors can be significantly affected by noisy data (Table 4).

| Noise | 1st eigenvector | 2nd eigenvector |
|---|---|---|
| sym 20 | $0.015 \pm 0.009$ | $0.043 \pm 0.021$ |
| sym 50 | $0.029 \pm 0.019$ | $0.078 \pm 0.044$ |
| sym 80 | $0.057 \pm 0.038$ | $0.135 \pm 0.052$ |

Table 4: Comparison of the perturbations of Eq. (1) ($\| \boldsymbol{u}\boldsymbol{u}^\top - \boldsymbol{v}\boldsymbol{v}^\top \|$) on CIFAR-10 with symmetric noise. The values in the table are written as mean (std) of the perturbations between u and v obtained for each class.

**Comparison to the mean vector.**   The mean vector can be a nice ad-hoc solution as a decision boundary. This is because the first eigenvector of the gram matrix and the mean vector of the cluster become similar under a low noise ratio. However, because the gram matrix of the cluster becomes larger in a high noise ratio scenario, naive averaging can cause a lot of perturbation. On the other side, because the first eigenvector arises from the principal component of the representation vectors, FINE is more robust to noisy representations so that it has less perturbation and provides better performance.

To support this explanation, we performed additional experiments by changing the anchor point with the first eigenvector and the mean vector. As Table 5 shows, the performance degradation occurs as the noise ratio increases by replacing the first eigenvector with the mean vector

| Noise | sym 20 | | sym 50 | | sym 80 | |
|---|---|---|---|---|---|---|
| | mean | eigen | mean | eigen | mean | eigen |
| Acc (%) | 90.32 | 91.42 | 86.03 | 87.20 | 67.78 | 71.55 |
| F-score | 0.8814 | 0.9217 | 0.4879 | 0.8626 | 0.6593 | 0.7339 |

Table 5: Comparison of test accuracies on the CIFAR-10 dataset.

### C.2  Detailed values for Figure 7

We provide the detailed values for Figure 7 in Table 6.

Table 6: Test accuracies (%) on CIFAR-10 and CIFAR-100 under different noisy types and fractions for noise-robust loss approaches. The average accuracies and standard deviations over three trials are reported.

| Dataset | CIFAR-10 | | | | CIFAR-100 | | | |
|---|---|---|---|---|---|---|---|---|
| Noisy Type | Sym | | | Asym | Sym | | | Asym |
| Noise Ratio | 20 | 50 | 80 | 40 | 20 | 50 | 80 | 40 |
| Standard | $87.0 \pm 0.1$ | $78.2 \pm 0.8$ | $53.8 \pm 1.0$ | $80.1 \pm 1.4$ | $58.7 \pm 0.3$ | $42.5 \pm 0.3$ | $18.1 \pm 0.8$ | $42.7 \pm 0.6$ |
| GCE | $89.8 \pm 0.2$ | $86.5 \pm 0.2$ | $64.1 \pm 1.4$ | $76.7 \pm 0.6$ | $66.8 \pm 0.4$ | $57.3 \pm 0.3$ | $29.2 \pm 0.7$ | $47.2 \pm 1.2$ |
| SCE* | $89.8 \pm 0.3$ | $84.7 \pm 0.3$ | $68.1 \pm 0.8$ | $82.5 \pm 0.5$ | $70.4 \pm 0.1$ | $48.8 \pm 1.3$ | $25.9 \pm 0.4$ | $48.4 \pm 0.9$ |
| ELR* | $91.2 \pm 0.1$ | $88.2 \pm 0.1$ | $72.9 \pm 0.6$ | $90.1 \pm 0.5$ | $74.2 \pm 0.2$ | $59.1 \pm 0.8$ | $29.8 \pm 0.6$ | $73.3 \pm 0.6$ |
| FINE | $91.0 \pm 0.1$ | $87.3 \pm 0.2$ | $69.4 \pm 1.1$ | $89.5 \pm 0.1$ | $70.3 \pm 0.2$ | $64.2 \pm 0.5$ | $25.6 \pm 1.2$ | $61.7 \pm 1.0$ |
| GCE + FINE | $91.4 \pm 0.1$ | $86.9 \pm 0.1$ | $75.3 \pm 1.2$ | $88.9 \pm 0.3$ | $70.5 \pm 0.1$ | $61.5 \pm 0.5$ | $37.0 \pm 2.1$ | $62.4 \pm 0.5$ |
| SCE + FINE | $90.4 \pm 0.2$ | $85.1 \pm 0.2$ | $70.5 \pm 0.8$ | $86.9 \pm 0.3$ | $70.9 \pm 0.3$ | $64.1 \pm 0.7$ | $29.9 \pm 0.8$ | $64.3 \pm 0.3$ |
| ELR + FINE | $91.5 \pm 0.1$ | $88.5 \pm 0.1$ | $74.7 \pm 0.5$ | $91.1 \pm 0.2$ | $74.9 \pm 0.2$ | $66.7 \pm 0.4$ | $32.5 \pm 0.5$ | $73.8 \pm 0.4$ |

## C.3 Hyperparameter sensitivity towards $\zeta$

We perform additional experiments; we report the test accuracy and f1-score on CIFAR-10 with a symmetric noise ratio of 80% across the value of the hyperparameter (Table 7). We can observe that the performance change is small in the acceptable range from 0.4 to 0.6.

| $\zeta$ | 0.4 | 0.45 | 0.5 | 0.55 | 0.6 |
|---|---|---|---|---|---|
| Acc (%) | 69.75 | 73.02 | 71.55 | 68.80 | 67.78 |
| F-score | 0.7270 | 0.7466 | 0.7339 | 0.7165 | 0.7016 |

Table 7: Sensitivity analysis for hyperparameter zeta on CIFAR-10 with symmetric noise 80%.

## C.4 Feature-dependent Label Noise

We additionally conducted experiments with our FINE methods on the feature-dependent noise labels dataset (noise rates of 20% and 40% by following the experimental settings of CORES [32]) (Table 8). To compare our FINE to CORES[2].

## C.5 Filtering Time Analysis

We compare the training times per one epoch of FINE with other filtering based methods, using a single Nvidia GeForce RTX 2080. We also report the computational time when the different number of data is used for eigen decomposition. We discover that there remain little difference as the number of instances is differently used for eigen decomposition. As Table 9 shows, the computational efficiency for FINE can be obtained without any degradation issues.

| Dataset | Noise | 0.2 | 0.4 |
|---|---|---|---|
| CIFAR-10 | CORES[2] | 89.50 | 82.84 |
| CIFAR-10 | FINE | 89.84 | 86.68 |
| CIFAR-100 | CORES[2] | 61.25 | 48.96 |
| CIFAR-100 | FINE | 62.21 | 47.81 |

Table 8: Comparison of test accuracies on clean datasets under feature-based label noise.

Table 9: Filtering Time Analysis on CIFAR-10 dataset

| DivideMix [25] | FINE | FINE using 1% dataset | F-Dividemix | F-Dividemix using 1% dataset |
|---|---|---|---|---|
| 2.2s | 20.1s | 1.1s | 40.2s | 2.1s |

## C.6  Other Real-world Noisy Datasets

We conduct additional experiments with our FINE method on the mini Webvision dataset for comparison with state-of-the-art methods (Table 10). In the comparison with CRUST [32], which is the state-of-the-art sample selection method, our method achieved 75.24% while CRUST achieved 72.40% on the test dataset of (mini) Webvision. Looking at the results, the difference between Dividemix [25] and F-Dividemix is marginal (Table 10). However, the reason for this is that we have to reduce the batch size due to the limitation of our current GPU, and we cannot do hyperparameter tuning (e.g. weight decay, learning rate). The final version will be able to run experiments by supplementing this issue, and it is expected that the performance will be improved.

| Method | Webvision | | ImageNet | |
|---|---|---|---|---|
| | top1 | top5 | top1 | top5 |
| CRUST [32] | 72.40 | 89.56 | 67.36 | 87.84 |
| FINE | 75.24 | 90.28 | 70.08 | 89.71 |
| DivideMix [25] | 77.32 | 91.64 | 75.20 | 90.84 |
| F-DivideMix | 77.28 | 91.44 | 75.20 | 91.28 |

Table 10: Comparison with state-of-the-art methods trained on (mini) WebVision dataset. Numbers denote top-1 (top-5) accuracy (%) on the WebVision validation set and the ImageNet ILSVRC12 validation set.