# OpenReview forum: "FINE Samples for Learning with Noisy Labels"
_NeurIPS.cc/2021/Conference — NeurIPS 2021 Poster_

### Official Review · Reviewer_VZE7 · 2021-07-05

**Rating:** 7
**Confidence:** 4

**Summary:**

This paper proposes a new method for learning with noisy labels. The proposed method belongs to the research direction of data-selection, and aims to better select the clean data out of noisy samples. The key component of the proposed method is a novel operation, named _filtering noisy labels via their eigenvectors_ (FINE). FINE first computes the covariance matrix of the feature representation for the noisy samples. Then the principal eigenvector is extracted from the covariance matrix via eigen decomposition. Afterwards, the similarity score between the feature vector of each sample and the eigenvector is computed. Finally, the computed similarly scores are used to fit a Gaussian Mixture Model, which predicts the probability that a sample is clean or noisy. The proposed FINE approach is simple and computationally feasible. Theoretically it is shown that FINE is able to approximate the eigenvector of clean data with guaranteed accuracy under a few reasonable assumptions. In practice, FINE is demonstrated to be effective by integrating it into three different kinds of label noise learning approaches. The experimental results show that FINE brings (much) performance gain over previous methods across several synthetic and real-world noisy datasets.

To summarize, the contributions of this paper include:

1. A new method, named FINE, is developed to better select the clean data out of noisy dataset. Theoretically it is shown that FINE is able to approximate the true eigenvector of clean data with guaranteed upperbound. Also, it is shown that the data selection accuracy of FINE has guaranteed lower bound in terms of precision and recall.

2. This paper also includes a few illustrative experiments to explain the characteristics of the proposed method. These experiments are helpful.

3. In practice, FINE is integrated into three different kinds of label noise learning approaches, and (significant) performance gain is observed over different datasets, involving the synthetic and real-world noisy data.

**Limitations And Societal Impact:**

The authors have addressed some of the limitations. Another limitation of this paper is that, the proposed method is subject to the degree of data imbalance.

This work does not have potential negative societal impact.

**Main Review:**

Overall, this paper is well motivated and has a clear organization. I think this paper is novel in terms of technical contribution and experimental evaluation:

1. The proposed data-selection method is reasonably designed, with theoretical analysis and insights. The proof the theory is complete and illustrates the key ideas (yet with some typos and unclear definitions, though).

2. Besides, to illustrate the characteristics of the proposed method, a series of experiments are conducted (on Page 6). These experiments are interesting, reasonably designed, and helpful in understanding the behavior of the method. In particular, I find Figure 3(a)(b) particularly useful and interesting. Also, the authors show that the proposed FINE can be efficiently computed via a simple approximate yet accurate strategy (i.e., using a subset of the noisy data). Such an experiment validation is useful since it shows the feasibility of FINE on large-scale datasets.

3. Finally, to show the effectiveness and generality of FINE, the paper investigates the performance gain by integrating FINE into three different kinds of noisy label learning methods. The experimental results on synthetic datasets are impressive. In particular, significant performance gain is achieved under high noise ratios (80% symmetric noise and 40% asymmetric noise).

Besides the above merits of this paper, it also has some weaknesses as follows:

1. The literature review is inadequate. In particular, this paper only mentions and discusses two directions for handling noisy labels: data selection and designing noise-robust loss. Another important direction, i.e., label correction, is ignored. Label correction tries to correct the wrong labels and thus is different from the noise-robust loss and data-selection. Some existing works along this direction include: Li _et al_ (ICCV 2017), Tanaka _et al_ (CVPR 2018), Yi _et al_ (CVPR 2019), Zheng _et al_ (ICML 2020).

&nbsp;&nbsp;&nbsp;&nbsp;&nbsp;&nbsp;&nbsp;&nbsp;Reference:

&nbsp;&nbsp;&nbsp;&nbsp;&nbsp;&nbsp;&nbsp;&nbsp;Li _et al_. Learning from noisy labels with distillation. ICCV, 2017.

&nbsp;&nbsp;&nbsp;&nbsp;&nbsp;&nbsp;&nbsp;&nbsp;Tanaka _et al_. Joint optimization framework for learning with noisy labels. CVPR, 2018.

&nbsp;&nbsp;&nbsp;&nbsp;&nbsp;&nbsp;&nbsp;&nbsp;Yi _et al_. Probabilistic end-to-end noise correction for learning with noisy labels. CVPR, 2019.

&nbsp;&nbsp;&nbsp;&nbsp;&nbsp;&nbsp;&nbsp;&nbsp;Zheng _et al_. Error-bounded correction of noisy labels. ICML, 2020.

2. The accuracy of the proposed method is heavily subject to the degree of data imbalance. More specifically, to achieve good precision and recall for data-selection, the number of clean samples (i.e., $N_+$) should be much larger than the number of noisy samples (i.e., $N_-$). This phenomenon can be observed from Line 13 of Algorithm 1, Definition 2, Figure 2, and Eq.(21)(22) in the Appendix. More specifically, when $N_- \geq N_+$, the precision and recall would be low, as indicated by the numerator of Eq.(21) and denominator of Eq.(22). This can also be visually inspected in Figure 2. For example, when $N_- \geq N_+$, the principal eigenvector would be skewed towards the noisy samples; as a result, the F-score will be largely hurt because GMM would mistakenly predict many noisy samples as clean. Therefore, only in cases where $N_- \leq N_+$ the proposed method would achieve a good performance. However, in practice this condition may not always hold.

3. While the theoretical results are informative, they are for the binary case. It is unclear if the results would be extended to multi-class scenario.

4. The Definition 2 is not very clear. In particular, "... if the representation $z$ labeled as the same true class belongs to the clean set" is a bit confusing. Does this sentence mean for all the samples whose noisy labels are the same with their underlying clean labels, they are successfully moved to the clean set? I think this sentence should be refined to make it clearer.

5. Figure 3(a)(b) are illustrative. However, it is better to also provide the heatmaps for ground-truth principal eigenvector. This is because in some cases when the computed $u$ is deviated from the ground-truth eigenvector, the heatmaps of $u$ would still appear as Figure 3(a)(b). By providing the heatmaps of ground-truth principal eigenvector, it is clearer to see if the heatmaps of computed $u$ are as bright as those of ground-truth eigenvector, and thereby check if the computed $u$ is accurate enough.

6. A minor concern: In Figure 3(c), although the perturbation values are small for low noise ratio (20%), they increase a lot for a few categories under high noise ratio (80%). Can we still say "FINE has small perturbation values even in a severe noise rate"?

7. For the combination of FINE and Co-teaching, does it mean both the two networks use FINE for data-section within a batch? And one network uses the selected data from another network for training itself? If so, does the batch size need to be large for effectively computing FINE score? Do the two networks still need warmup? Overall I find the description "we train one model with extracted samples by conducting FINE on another model" is unclear. Further clarification is necessary.

8. In Section 4.2.3, it is unclear how to combine FINE with noise-robust loss functions. No descriptions are provided. Does it mean Algorithm 2 is performed here, but the loss function $\mathcal{L}$ is replaced with the noise-robust loss (such as GCE)?

9. In Table 3, it is reported that DivideMix [26] achieves 74.30% test accuracy on Clothing1M. However, after checking the paper of DivideMix, I find it achieved 74.76% as reported in the original paper [26]. Is this a mistake? Overall the performance gain of FINE is marginal on the Clothing1M dataset.

10. There are some typos and grammatical errors in the paper. Besides, some descriptions are unclear. The writing should be further improved. For example:

-  Line 37-38: "loss value between ... and its noisy label". I think this description is confusing. A better one could simply be "the loss value of the noisy classifier". Line 39, "the magnitude", where "the" should be removed. Line 53: "and separates" -> ", and separates" to break the long sentence. Line 92: ", then" -> ", and then". Line 113: "Base on this" -> "Based on this". Line 152: "presented along" -> "presented with". "explore" -> "exploit"? The description "maximize the separation of alignment clusterability" is confusing, and need to be refined. Line 201, the "$v$" should be italic. Line 241: "F-scores of FINE becomes" -> "F-scores of FINE become". "the access of noise rate" -> "the access to noise rate". "and10k"->"and 10k". "methods belong to this" -> "methods belonging to this". "clean and noisy data is" -> "clean and noisy data are".

- For Lemma 3, I think it is more commonly to call it "David-Kahan sin($\theta$) Theorem".

- One typo in Eq.(20). $(\frac{1}{N_+} + \frac{1}{N_+})$ should be $(\frac{1}{N_-} + \frac{1}{N_+})$.

- Line 659, "extension on" -> "extension to".

- Line 699, "there remain little difference as the number of instances is differently used for eigen decomposition". This is misleading, because the time cost changes a lot from 20.1s to 1.1s, when switching from "FINE" to "FINE using 1% dataset".

- Overall, **the writing of this paper should be further improved**.


To summarize, although this paper has several issues, especially those about unclear descriptions and typos, this paper still has a few interesting points. In terms of data-selection, it achieves better performance than the previous works (e.g., CRUST, TopoFilter, etc.) on synthetic datasets. Therefore, I think this paper would potentially meet the standard of NeurIPS after addressing my concerns, and could be **conditionally** accepted.

**Time Spent Reviewing:**

18

---

> ### Author Response · Authors · 2021-08-10
> **Response to Reviewer VZE7**
>
> Thank you for your careful review of our paper and your insightful and constructive comments. We have addressed your comments and updated our manuscript accordingly. Please find our detailed answers below.
>
> **Q1) Complement literature review**
>
> A1) Thank you for your comment on this part which is missing from our manuscript. We have added a literature review regarding label correction. Please find the text below.
>
> "Veit et al. [1] provided a semi-supervised learning framework that facilitates small sets of clean instances and an additional label cleaning network to correct the massive sets of noisy labels.  Li et al. [2] suggested a distillation framework that applies knowledge graphs to correct noisy labels. Tanaka et al. [3] proposed a joint optimization framework that optimizes the network parameters and class labels using an alternative strategy. Lee et al. [4] introduced CleanNet, which learns a class-embedding vector and a query-embedding vector with a similarity matching constraint to identify the noise instances that have less similar embedding vectors with their class-embedding vectors. Li et al. [5] provided a probabilistic learning framework that utilizes label distribution updated by back-propagation to correct the noisy labels. Zheng et al. [6] showed that noisy classifiers and the purity of labels are highly correlated and suggested a label correction algorithm by detecting noise instances."
>
> **Q2) Degree of data imbalance ($N_{+}$ vs. $N_{-}$)**
>
> A2) For the theoretical analysis (Theorem 1), we only consider the binary classification referred to in [7].  Hence, it is unnatural that $N_{-}$ is larger than $N_{+}$. In this setting, even in practice, we believe that this condition may hold. Real-world cases are generally not binary classification problems. Then, the total number of noise instances can be larger than the number of clean instances, whereas the number of noise instances whose true class is i is smaller than clear instances for every class i. However, we can still use the first eigenvector to generate clean data. We explain how to expand Theorem 1 to multiple cases in answer to the next question.
>
> **Q3) Expanding theorem 1 to multiple classification cases.**
>
> A3) We can show that Theorem1 is easily expanded to multiple classification cases by defining the vector of noisy instances with true class i as w_{i} instead of defining a single noisy vector w. In this scenario, the principal eigenvector is from the clean data; the perturbation between this principal eigenvector and each noise class can be obtained through Theorem 1, and if the upper bound is obtained from the sum of all perturbations, it can be extended to a multi-classification problem.
>
> **Q4) Refinement on the definition 2**
>
> A4) We will modify definition 2 more clear as follows:  "... if all the features labeled as the same true class belongs to the clean set.". We call this alignment clusterability if all really clean data are well classified into a clean set.
>
> **Q5) Heatmap for ground-truth principal eigenvector**
>
> A5) Thank you for your suggestion. We will revise Figures 3(a) and (b) by adding the ground-truth principal eigenvector.
>
> **Q6) A minor concern; the magnitude of the perturbation.**
>
> A6) In Figure 3(c), the perturbation values are relatively small for low noise ratios (20%, approximately 0.02 for airplanes) and relatively large for high noise ratios (80%, approximately 0.11 for airplanes). However, because the perturbation (i.e., $||uu^{\intercal} - vv^{\intercal }||$) is the sine function value (see Lemma 1. in Appendix) of the angle between u (whole data's eigenvector) and v (clean data's eigenvector), we can obtain angles close to \pi/2 in both the low noise ratio case and high noise ratio case (arcsine(0.02) = 1.15', arcsine(0.11) = 6.40'). Hence, we can still say "FINE has small perturbation values even with severe noise."
>
> **Q7) Adding the description about the combination of FINE and Co-teaching**
>
> Thank you for your constructive comment. We have revised the explanation more clearly. To combine FINE with Co-teaching [8], we borrowed the main idea of Co-teaching [8] in which one network uses the selected data from another network for the training itself. However, we did not perform data selection within a batch. Before the parameter updating step, we conduct FINE on the whole data for training one network by utilizing another network and vice versa. Hence, it may be that the lengths of the two datasets were different.
>
> **Q8) Adding the description about the collaboration with noise-robust loss functions.**
>
> A8) We conducted experiments by replacing the cross-entropy loss with the noise-robust loss (GCE, SCE, ELR) in Section 4.2.3. We will provide further detail in the supplementary material (Appendix B.3).
>
> **Q9) The performance of Dividemix in paper**
>
> A9) The accuracy of 74.30% in this study was reproduced by conducting experiments using the official code of Dividemix. We tried our best to reproduce Dividemix's original performance; however, it was difficult to achieve the best performance described in the original paper. Instead, we performed experiments using F-DivideMix and DivideMix on the same setup and obtained higher accuracy with our F-DivideMix. Moreover, to compensate for the marginal performance improvement in the Clothing1M dataset, we performed additional experiments on the Webvision dataset and showed extraordinary improvement compared to the existing sample selection methods and semi-supervised based approach.
>
> **Q10) Typos and grammatical errors, unclear descriptions**
>
> A10) Thank you for your attention to detail. To prevent such errors and confusion, we will revise the paper and clarify our unclear descriptions and improve our overall writing overall.
>
> **References**
>
> [1] Veit *et al.* Learning from noisy large-scale datasets with minimal supervision, CVPR, 2018.
>
> [2] Li *et al*. Learning from noisy labels with distillation. ICCV, 2017.
>
> [3] Tanaka *et al*. Joint optimization framework for learning with noisy labels. CVPR, 2018.
>
> [4] Lee *et al.* CleanNet: Transfer learning for scalable image classifier training with label noise. CVPR, 2018.
>
> [5] Yi *et al*. Probabilistic end-to-end noise correction for learning with noisy labels. CVPR, 2019.
>
> [6] Zheng *et al*. Error-bounded correction of noisy labels. ICML, 2020.
>
> [7] A Topological Filter for Learning with Label Noise, NeurIPS 2020.
>
> [8] Co-teaching: Robust Training of Deep Neural Networks with Extremely Noisy Labels, NIPS 2018.

---

> > ### Comment · Reviewer_VZE7 · 2021-08-15
> > **The authors have addressed most of my concerns**
> >
> > Thank you for your detailed response. I think the authors have adequately addressed most of my concerns.
> >
> > There still remains one concern, i.e., my second concern regarding the degree of data imbalance. Specifically, why for a binary classification task "it is unnatural that $N_-$ is larger than $N_+$"? In my view, this could happen in principle. For example, for class $i$, it is possible that 90% of the clean data are flipped to other classes, and only the remaining 10% data are clean. In this scenario, I don't see why "the number of noise instances whose true class is $i$ is smaller than clear instances for every class $i$". Could you please add more details regarding this?
> >
> > Finally, I hope the authors would carefully refine the paper writing to make it easier to understand.
> >
> > I am satisfied with the responses with respect to my other concerns.

---

> > > ### Author Response · Authors · 2021-08-26
> > > **Replies for your valuable review**
> > >
> > >
> > > Thank you for your thoughtful reading of our responses and great suggestions. In our opinion, your question about data imbalance in binary classification can be explained largely in two cases.
> > >
> > > Question) Concern about $N_{-} < N_{+}$ : "For class $i$, while the 90% of the clean data are flipped to other classes, and only the remaining 10% data are clean"
> > >
> > >
> > > **Case 1)** Both classes are flipped into each class.
> > > In this case, each class has much more noise data than clean data. In other words, in binary classification, mislabeled data is much more major than data with ground-truth labels. (e.g., for class 0, only 10% data are correctly labelled and 90% have ground-truth label 1.)In this scenario, we wanted to point out that the situation in which ground-truth is not major is unnatural. In this example, we think that modifying the label 10% of data to the opposite class would be better. Consider a symmetrical example. If the problem of the existing situation is solved, the methodology will also attempt to modify the label if the clean is 90% and the noise is 10%. This error can occur, so it is unnatural. A similar assumption has been held in other recent LNL methods [1, 2, 3].
> > >
> > > [1] A Topological Filter for Learning with Label Noise, NeurIPS 2020.
> > > [2] Error-Bounded Correction of Noisy Labels, ICML 2020.
> > > [3] Learning with Instance-dependent Label Noise: A Sample Sieve Approach, ICLR 2021
> > >
> > > **Case 2)** only one class is flipped.
> > > This is an asymmetric case. When there are 100 data with ground-truth classes of 0 and 1, respectively, 90 data with class 0 are incorrectly labeled as class 1. In this case, class 0 has 10 data, class 1 has 190 data, and each class still has clean data majority.

---

> > > > ### Comment · Reviewer_VZE7 · 2021-09-02
> > > > **Thank you for the response**
> > > >
> > > > Thank you for your response. To some degree, I agree with the claim that "the situation in which ground-truth is not major is unnatural". However, in my view, this unnatural situation could still happen in practice; for example, the labels could be maliciously flipped by some attachers.
> > > >
> > > > I think the authors have addressed all of my concerns. Thanks a lot.

---

### Official Review · Reviewer_utGV · 2021-07-13

**Rating:** 7
**Confidence:** 4

**Summary:**

This paper proposes FINE detector to detect samples with noisy labels. Specifically, authors first construct data covariance matrix by using the latent representation of each training sample, then perform eigen-decomposition on the data covariance matrix to generate the first eigenvector for each class. The authors state that the representation of clean samples have larger alignment score with eigenvector than noisy ones. The authors also provide theoretical results to analyze the robustness of FINE detector and shows this detector can be applied to multiple approaches including sample-selection, semi-supervised learning and collaboration with noise-robust loss functions. Experiments are conducted on CIFAR-10, CIFAR-100 and Clothing1M.

**Limitations And Societal Impact:**

Authors have adequately addressed the limitations and potential negative societal impact of their work.

**Main Review:**

***Strength***

**1:** The noisy label detector is simple and novel. Necessary theoretical results are also provided to verify its robustness.

**2:** I especially like the analysis part in Figure 3. It coincides with the theorem to help better understand the FINE detector.

**3:** Experimental results on three applications are impressive. It shows that FINE can be combined with other approaches to further improve network performance.

***Concerns***

**1:** Since the noise rate is unknown, I am worried that FINE is sensitive to the hyper-parameter $\zeta$ in Algorithm 1.  Authors should clarify the value of $\zeta$  for each experiment setting and perform sensitivity analysis on $\zeta$.

**2:** The noise types in the paper are symmetric and asymmetric label noise which are assumed to be feature-independent. I am wondering how FINE performs on a more challenging setting such as feature-dependent label noise.

**3:** FINE only uses the first eigenvector. If it is possible to use multiple eigenvectors to calculate alignment scores. Is the result better than using only one single eigenvector?

**Time Spent Reviewing:**

10

---

> ### Author Response · Authors · 2021-08-10
> **Response to Reviewer utGV**
>
> Thank you for your careful review of our paper and your insightful and constructive comments. We have addressed your comments and updated our manuscript accordingly. Please find our detailed answers below.
>
> **Q1) Hyper-parameter sensitivity towards $\zeta$**
>
> A1) To answer this question, we performed additional experiments; we reported the test accuracy and f1-score on CIFAR-10 with a symmetric noise ratio of 80% across the value of the hyperparameter $\zeta$ (Table1). We can observe that the performance change is small in the acceptable range from 0.4 to 0.6.
>
> |   $\zeta$  |   0.4  |  0.45  |   0.5  |  0.55  |   0.6  |
> |:-------:|:------:|:------:|:------:|:------:|:------:|
> | Acc (%) |  69.75 |  73.02 |  71.55 |  68.80 |  67.78 |
> | F-score | 0.7270 | 0.7466 | 0.7339 | 0.7165 | 0.7016 |
>
> Table1. Sensitivity analysis for hyperparameter zeta on CIFAR-10 with symmetric noise 80%
>
> **Q2) Feature-dependent label noise**
>
> A2) Thank you for your suggestion. We additionally conducted experiments with our FINE methods on the feature-dependent noise labels dataset (noise rates of 20% and 40% by following the experimental settings of CORES$^2$ [1]) (Table2). To compare our FINE to CORES$^2$.
>
> |  Dataset  |   Noise   |  0.2  |  0.4  |
> |:---------:|:---------:|:-----:|:-----:|
> |  CIFAR-10 | CORES$^2$ | 89.50 | 82.84 |
> |  CIFAR-10 |    FINE   | 89.84 | 86.68 |
> | CIFAR-100 | CORES$^2$ | 61.25 | 48.96 |
> | CIFAR-100 |    FINE   | 62.21 | 47.81 |
>
> Table2. Comparison of test accuracies on clean datasets under feature-based label noise.
>
> **Q3) How about using multiple eigenvectors?**
>
> A3) Thank you for your insightful comment. We provide robustness by means of the way the first eigenvector is robust to noisy vectors so that FINE can fix the noisy classifier by using segregated clean data. Unlike the first eigenvector, the other eigenvectors can be significantly affected by noisy data. Please refer to the following table.
>
> |  Noise  | 1st eigenvector | 2nd eigenvector |
> |:-------:|:---------------:|:---------------:|
> | sym 20% |  0.015 (0.009)  |  0.043 (0.021)  |
> | sym 50% |  0.029 (0.019)  |  0.078 (0.044)  |
> | sym 80% |  0.057 (0.038)  |  0.135 (0.052)  |
>
> Table3. Comparison of the perturbations of Eq. (1) ($||uu^{\intercal} - vv^{\intercal }||$) on CIFAR-10 with symmetric noise. The values in the table are written as mean (std) of the perturbations between u and v obtained for each class.
>
> An example of using multiple eigenvectors is the mean vector. We provide detailed experimental results in Table 4. As shown in the results, the mean vector degrades performance compared to the first eigenvector.
>
> |  noise  |   0.2  |        |   0.5  |        |   0.8  |        |
> |:-------:|:------:|:------:|:------:|:------:|:------:|:------:|
> |         |  mean  |  eigen |  mean  |  eigen |  mean  |  eigen |
> | Acc (%) |  90.32 |  91.42 |  86.03 |  87.20 |  67.78 |  71.55 |
> | F-score | 0.8814 | 0.9217 | 0.8479 | 0.8626 | 0.6593 | 0.7339 |
>
> Table 4. Comparison of test accuracies on the CIFAR-10 dataset.
>
> **References**
>
> [1] Learning with Instance-Dependent Label Noise: A Sample Sieve Approach, ICLR 2021.

---

### Official Review · Reviewer_EK8d · 2021-07-15

**Rating:** 7
**Confidence:** 4

**Summary:**

This paper is about learning with noisy labels. The authors propose a method which selects clean and noisy examples by first computing a covariance matrix (for each class separately) based on the learned representations. They then compute the first eigenvector corresponding to the largest eigenvalue from the covariance matrix. The "cleanness" measurement then becomes the inner product between the original representation and the eigenvector. The authors show multiple applications of their approach, such as sample selection, semi-supervised learning etc., and show that their method consistently improves the classification accuracy.

**Limitations And Societal Impact:**

Yes

**Main Review:**

This paper proposes a simple but effective method based on the eigenvectors of representations. As far as I know, this is a novel approach, and is also supported by theoretical analysis in the paper. The authors also show that it can scale up to the large datasets by computing the eigenvectors on a smaller subset.

The paper is also well written and easy to understand. I also appreciate the fact that it can be easily combined with various existing learning with noisy labels approaches to further improve the accuracy. Overall, it's a nice paper to read with convincing results. However, I have some minor comments:

- I think the paper can benefit from one baseline to show why the covariance matrix and eigenvectors are needed. More specifically, instead of computing the first eigenvectors, one can compute the class mean vector v_c, where v_c = 1/n_c \sum z_i where all z_i belongs to class C, and n_c is the number of elements in class C. One can then compute the FINE score based on this vector, i.e. f_i = <v_c, z_i>. This comparison would make it more convincing to show why we need to compute the eigenvectors.

- The proposed method brings minimal improvements in the real-world dataset experiments. Can the authors comment on why this is the case? Have they tried other real-world noisy datasets, e.g. Webvision, Food-101N?

- Nitpick: I was initially confused by the terminology "covariance matrix" used throughout the paper. The confusion was mainly due to the lack of notation in the paper, i.e. the reader does not know if z_i is considered as a dx1 vector or 1xd. I assumed that it was it was dx1, which would make the "covariance" matrix dxd, as in PCA. However, after looking at the code (in supplementary material), it seems like that the "covariance matrix" is actually nxn, where n is the number of vectors in a class. Can the authors clarify this? If it is indeed nxn, perhaps calling it a Gram matrix would be better?

**Time Spent Reviewing:**

1

---

> ### Author Response · Authors · 2021-08-10
> **Response Reviewer EK8d**
>
> Thank you for your careful review of our paper and your insightful and constructive comments. We have addressed your comments and updated our manuscript accordingly. Please find our detailed answers below.
>
> **Q1) More supports for the replacement with the first eigenvector in the mean vector or other eigenvectors.**
>
> A1) Thank you for your insightful comment. As you suggested, the mean vector can be a nice ad-hoc solution as a decision boundary. This is because the first eigenvector of the gram matrix and the mean vector of the cluster become similar under a low noise ratio. However, because the covariance of the cluster becomes larger in a high noise ratio scenario, naive averaging can cause a lot of perturbation. On the other side, because the first eigenvector arises from the principal component of the representation vectors, FINE is more robust to noisy representations so that it has less perturbation and provides better performance.
>
> To support this explanation, we performed additional experiments by changing the anchor point with the first eigenvector and the mean vector. As Table1 shows, the performance degradation occurs as the noise ratio increases by replacing the first eigenvector with the mean vector.
>
> |  noise  |   0.2  |        |   0.5  |        |   0.8  |        |
> |:-------:|:------:|:------:|:------:|:------:|:------:|:------:|
> |         |  mean  |  eigen |  mean  |  eigen |  mean  |  eigen |
> | Acc (%) |  90.32 |  91.42 |  86.03 |  87.20 |  67.78 |  71.55 |
> | F-score | 0.8814 | 0.9217 | 0.8479 | 0.8626 | 0.6593 | 0.7339 |
>
> Table 1. Comparison of test accuracies on the CIFAR-10 dataset.
>
> **Q2) Other real-world noisy datasets**
>
> A2) To answer this question, we conducted additional experiments with our FINE method on the mini Webvision dataset for comparison with state-of-the-art methods. In the comparison with CRUST [1], which is the state-of-the-art sample selection method, our method achieved 75.24% while CRUST achieved 72.40% on the test dataset of (mini) Webvision. For this rebuttal, we also tested the SSL approach F-dividemix on the webvision dataset.
>
> |    Method   | Webvision |       | ImageNet |       |
> |:-----------:|:---------:|:-----:|:--------:|:-----:|
> |             |   top 1   | top 5 |   top 1  | top 5 |
> |    CRUST    |   72.40   | 89.56 |   67.36  | 87.84 |
> |     FINE    |   75.24   | 90.28 |   70.08  | 89.71 |
> |  DivideMix  |   77.32   | 91.64 |   75.20  | 90.84 |
> | F-DivideMix |   77.28   | 91.44 |   75.20  | 91.28 |
>
> Table2. Comparison with state-of-the-art methods trained on (mini) WebVision dataset. Numbers denote top-1 (top-5) accuracy (%) on the WebVision validation set and the ImageNet ILSVRC12 validation set.
>
> Looking at the results, the difference between Dividemix [2] and F-Dividemix is marginal (Table2). However, the reason for this is that we have to reduce the batch size due to the limitation of our current GPU, and we cannot do hyperparameter tuning (e.g. weight decay, learning rate). The final version will be able to run experiments by supplementing this issue, and it is expected that the performance will be improved.
>
> **Q3) Nitpick for the terminology "covariance matrix"**
>
> A3) Thank you for your comment regarding the lack of notation. We will replace the "covariance matrix" with "gram matrix" and clarify the notation.
>
> **References**
>
> [1] Coresets for Robust Training of Neural Networks against Noisy Labels, NeurIPS 2020.
>
> [2] DIVIDEMIX: LEARNING WITH NOISY LABELS AS SEMI-SUPERVISED LEARNING, ICLR 2020.

---

### Official Review · Reviewer_YKHM · 2021-07-27

**Rating:** 6
**Confidence:** 4

**Summary:**

Deep neural networks become inefficient when the datasets contain noisy class labels. There are two types of robust techniques that are applied in the presence of the noisy labels: a) noise-robust loss functions; b) removing noisy data by detecting them. Most existing detection methods are dependent on the loss values, while such losses may be biased by corrupted classifier. This paper proposes a method that attempts to alleviate this issue by extracting key information from representations, but without using explicit knowledge of the noise rates.

The paper proposes a new method for filtering noisy data. It focuses on each data’s latent representation dynamics and measure the alignment between the latent distribution and each representation using the eigen decomposition of the data covariance matrix. This framework is coined as filtering noisy instances via their eigenvectors (FINE). It provides a detector with derivative-free simple methods with some theoretical guarantees. It provides a provable evidence that FINE allows meaningful decision boundary made by eigenvectors in latent space. It supports the theoretical analysis with various experimental results regarding the characteristics of the principal components extracted by the FINE detector.

It proposes three applications of the FINE: sample-selection approach, semi-supervised learning approach, and collaboration with noise-robust loss functions. It empirically validates that a sample-selection learning with FINE provides consistently more superior detection quality and higher test accuracy than other existing alternative methods such as Co-teaching families, TopoFilter, and CRUST. Further, experimental results show that the proposed methods consistently outperform corresponding baselines for all three applications on various benchmark datasets.

**Limitations And Societal Impact:**

Yes.

**Main Review:**

The proposed technique for detecting noisy data based on the degree of alignment between the representations and the eigenvector of the representations' covariance matrices for all classes is new and meaningful. The theoretical guarantees given in Theorem 1 are not very innovative but nonethless help understand the proposed approach. Further, numerical results for the case of "Sample-selection" shows that the proposed FINE approach consistently outperforms the baseline existing approaches for all the symmetric/assymetric noise settings on CIFAR-10 and CIFAR-100 datasets. These numerical results are the strongest part of the paper. Experiments for the setting of Semi-supervised learning show little gain over the baseline approaches. For all the numerical experiments, authors do compare their approach with multiple state-of-the-art competitive approaches.

The major limitation is that theoretical contribution of the paper is not very innovative and does not take into account the fact that the representations are learned dynamically using a noisy dataset. So, the original claim of the paper that their approach is different from the existing methods as in the existing methods are biased by the corrupted classifier is not true. Their proposed approach FINE suffers from the same issue as the representations that are used to compute the degree of alignment are latent representations of a corrupted classifier (a classifier leanred using noisy dataset).

**Time Spent Reviewing:**

6

---

> ### Author Response · Authors · 2021-08-10
> **Response to Reviewer YKHM**
>
> Thank you for your careful review of our paper and your insightful and constructive comments. We have addressed your comments and updated our manuscript accordingly. Please find our detailed answers below.
>
> **Q1) Does FINE suffer the same issue by using biased representations for computing the degree of alignment?**
>
> A1) While FINE uses biased representations, we provide robustness regarding how the first eigenvector is robust to noisy vectors so that FINE can fix the noisy classifier by using the segregated clean data. Our theorem guarantees the degree of the robustness bounds. In this respect, we think our FINE method is a simple yet effective novel method with theoretical guarantees. This is because FINE does the best job of filtering noise within the theoretical bounds. We additionally explain the vital role of the first eigenvector compared to other eigenvectors and the mean vector. We think these results can be an answer to you. Please refer to the followings.
>
> **Compared to other eigenvectors**
> We provide robustness by means of the way the first eigenvector is robust to noisy vectors so that FINE can fix the noisy classifier by using segregated clean data. Unlike the first eigenvector, the other eigenvectors can be significantly affected by noisy data. Please refer to the following table.
>
> |  Noise  | 1st eigenvector | 2nd eigenvector |
> |:-------:|:---------------:|:---------------:|
> | sym 20% |  0.015 (0.009)  |  0.043 (0.021)  |
> | sym 50% |  0.029 (0.019)  |  0.078 (0.044)  |
> | sym 80% |  0.057 (0.038)  |  0.135 (0.052)  |
>
> Table1. Comparison of the perturbations of Eq. (1) ($||uu^{\intercal} - vv^{\intercal }||$) on CIFAR-10 with symmetric noise. The values in the table are written as mean (std) of the perturbations between u and v obtained for each class.
>
>
> **Compared to the mean vector**
> The mean vector can be a nice ad-hoc solution as a decision boundary. This is because the first eigenvector of the gram matrix and the mean vector of the cluster become similar under a low noise ratio. However, because the covariance of the cluster becomes larger in a high noise ratio scenario, naive averaging can cause a lot of perturbation. On the other side, because the first eigenvector arises from the principal component of the representation vectors, FINE is more robust to noisy representations so that it has less perturbation and provides better performance.
>
> To support this explanation, we performed additional experiments by changing the anchor point with the first eigenvector and the mean vector. As Table2 shows, the performance degradation occurs as the noise ratio increases by replacing the first eigenvector with the mean vector.
>
> |  noise  |   0.2  |        |   0.5  |        |   0.8  |        |
> |:-------:|:------:|:------:|:------:|:------:|:------:|:------:|
> |         |  mean  |  eigen |  mean  |  eigen |  mean  |  eigen |
> | Acc (%) |  90.32 |  91.42 |  86.03 |  87.20 |  67.78 |  71.55 |
> | F-score | 0.8814 | 0.9217 | 0.8479 | 0.8626 | 0.6593 | 0.7339 |
>
> Table2. Comparison of test accuracies on the CIFAR-10 dataset.

---

### Author Response · Authors · 2021-08-10
**Summary of Reviews to all and AC**

We really thank all the reviewers (@**YKHM, @EK8d, @utGV, @VZE7**) for their thoughtful feedback. We are encouraged to find our motivation and ideas to be novel in terms of technical contribution and experimental evaluation (Reviewers **@YKHM, @EK8d, @utGV, @VZE7**) and theoretically guaranteed (Reviewers **@YKHM, @EK8d, @utGV, @VZE7**), and our results significant (Reviewers **@YKHM, @EK8d, @utGV, @VZE7**). We are glad all reviewers found our claims to be sufficiently supported by theoretical analysis and experimental results and technically sound (Reviewers **@YKHM, @EK8d, @utGV, @VZE7**). We are pleased that all reviewers feel our paper is well-organized, clearly written, and easy to understand (Reviewers **@YKHM, @EK8d, @utGV, @VZE7**).

We are pleased that reviewers identified that our new method, named filtering noisy labels via their eigenvectors (FINE), can better select the clean data from the noisy dataset. FINE first computes the covariance matrix of the latent representation (pre-logit) for the noisy samples. Then, the principal eigenvector is extracted from the covariance matrix via eigendecomposition. Afterward, the score between the feature vector of each sample and the eigenvector is used to fit a Gaussian Mixture Model, which predicts the probability that a sample is clean or noisy. The proposed FINE approach is simple and computationally feasible (Reviewer **@VZE7**). In addition, it can scale up the large datasets by computing the eigenvectors of a smaller dataset (Reviewer **@EK8d**).  We clearly provide the experimental results of three applications: a sample-selection approach, a semi-supervised learning approach, and a collaboration with a noise-robust loss function. The results show that FINE can be combined with other approaches to further improve network performance over different datasets, involving the synthetic and real-world noisy data (Reviewers **@YKHM, @EK8d, @utGV, @VZE7**). We provide technical analysis regarding the validity of our theorem for FINE in Figure 3 (Reviewer **@utGV**). Our code can be accessible via our attached supplement file, and after de-anonymization we will make it available through GitHub. We answer the following specific questions and incorporate all the feedback in the final version of the manuscript.

We believe that FINE can be applied to commercial models to detect noisy data or train itself to be robust to noisy data. Because noisy data can exist everywhere in the dataset, the detection of noisy data and training of the robust model is becoming increasingly important. FINE can be a nice solution for this real-world challenge and gives of social significance.

---

### Public Comment · Authors · 2022-06-28
**[Modification of Paper in Proceedings & OpenReview**

[Important] To all..!!

In the paper, some critical typo happens on page 4..!!

The Definition1 part is erased in the process of delivering the camera-ready version..!
(Definition 1 was well-written before the camera-ready version, but something wrong happens - maybe our mistakes..)

You can find the perfect version in arxiv
link: https://arxiv.org/abs/2102.11628

---

### Decision · Program_Chairs · 2021-09-27

**Decision:**

Accept (Poster)

**Comment:**

The paper proposes a method for identifying noisy examples by looking at their alignment with top eigenvector of the class specific feature covariance matrix. Reviewers have appreciated the simplicity of the method. Author responses were able to address the reviewers' concerns well and the paper is suitable to appear at the conference.